# Demonstrating the capacity of a Path-Based variational inference formulation for robust hidden Markov modeling of complex and noisy binary trees

## Abstract

Binary tree structures are prevalent across multiple fields, such as procedural modeling, genomics, and image processing. Hidden Markov models (HMMs) provide compact and interpretable representations for these complex and fractal structures. However, current de-facto inference methods involve complex iterations over all sub-trees, implementations that are domain-specific, and lack a unified open-source solution. This study explores a novel **'paths-of-bifurcations'** inference approach to fit hidden Markov parameters on binary trees, compatible with popular modeling packages. Key contributions include: (1) demonstration of procedural modeling for creating a sandbox of synthetic trees for experimentation; (2) comprehensive performance evaluations of our inference procedure on synthetic benchmark trees addressing various challenges: heterogeneity of branch emission distributions, low probability states, small data regimes, and noisy observational data; and (3) a practical application to a medical image dataset. The latter showcases the method's ability to reveal insights into branching rules governing the human airway system, with potential implications in disease characterization, airflow analysis, and particle deposition studies. This research provides a step toward robust, scalable and user-friendly generative modeling of binary tree structures with broad interdisciplinary implications.

## 1 Introduction

Binary tree structures are pervasive in botanics, genomics, and medical imaging for airways and vessels. Modeling of plant architectures is often viewed as repetitive branching rules into lateral shoots (Stava et al., 2014; Durand et al., 2005; Lieb et al., 2022) with procedural modeling. Cell lineages in genomics are modeled through the repetitive division of parent cells into two daughter cells with different molecular and phenotypic characteristics (Mohammadi et al., 2022; Biesinger et al., 2013). In signal processing, wavelet-based techniques have enabled robust denoising and edge localization by modeling wavelet coefficients across scales into Markov trees (Durand et al., 2004; Crouse et al., 1998). In medicine, airway and vascular structures also exhibit similar branching dynamics during embryonic development (Metzger et al., 2008; Abbasi & Boozarjomehry, 2021) and have the form of 3-D graph-encoded trees when viewed on CT, MRI, and other imaging modalities (Vos et al., 2023).

Fast and scalable inference of a generative process governing such tree structures is key for classifying trees into categories. In medicine, the characterization of airway and vascular systems is essential for tasks such as deciphering disease mechanisms, detecting early bio-markers, determining individual susceptibility for personalized medicine, and facilitating the creation of synthetic digital twins for airflow and particle deposition simulations.

However, state-of-the-art techniques, which include measuring topological distances between partial airway trees (Wysoczanski et al., 2021) or identifying anomalies within vascular trees (Vos et al., 2023) are discriminative. Hence, they do not tackle the encoding of the fundamental generative processes that govern the architecture of these biological systems. Manual inspection of growing airway structures in animal models has led to significant advances in understanding airway morphology and the impact of factors such as genetic influences (Metzger et al., 2008). However, such

generative modeling still needs to be explored in human subjects and with rigorous mathematical modeling formulations (eg. by defining and controlling the modeling variables).

Hidden Markov models (HMMs) can compactly represent iterative branching structures (independently of tree size). Full inference of tree-HMMs assigns each node in a tree with a state and infers state transition rules and emission distributions on state variables. Current inference frameworks involve complex iterations over all sub-trees. Durand et al. (2005) have demonstrated tree-HMM inference on botanical datasets; Mohammadi et al. (2022) demonstrated a pipeline tailored to cell lineage analysis, while Biesinger et al. (2013) have focused on chromatin state mapping. However, a unified, scalable, open-source framework for tree-structured data analysis has yet to be established.

This paper aims to reduce the complexity of analyzing 3-D graph-encoded tree structures. We examine the capabilities of a simple **'paths-of-bifurcations'** approximation for inferring HMM parameters from observed 3-D binary tree structures. We model this approximation by treating each bifurcation in the tree as a node with an assigned Markov state. We split trees into 'independent' paths of bifurcations from the root to each leaf node, assuming no correlation between paths during inference. This approximation enables the straightforward application of Rabiner (1989)'s technique for analyzing time-series data with HMMs. It also enables us to leverage the publicly-available GPU-accelerated Pyro package for scalable variational inference (Bingham et al., 2019).

We generate 3-D synthetic benchmark trees, via procedural modeling, to validate the ability of our 'paths-of-bifurcations' approximation to accurately infer HMM parameters across a large range of configurations: constrained tree structures, heterogeneous bifurcation rules, low probability rules, and observational noise. By doing this, we aim to extend access to tree-HMM modeling for a broad community. We calibrated our synthetic data on a use-case in medical imaging to characterize the branching patterns in human airway trees segmented on CT images. Code to experiment with synthetic benchmark trees and run HMM inference is available at: `https://anonymous.4open.science/r/paths-of-bifurcations-hmm-3BF4`.

## 2 MATERIALS AND METHODS

### 2.1 PROBABILISTIC PROCEDURAL MODELING & SYNTHETIC DATA GENERATION

A procedural model involves the repetitive application of a small set of rules that control local behavior. We use the Python package Lpy (Boudon et al., 2012) for procedural modeling of L-systems, a parallel string rewriting system used in graphics design (Prusinkiewicz & Lindenmayer, 2012), to implement our HMM generative model (Section 2.1.1) and generate synthetic 3-D tree-like structures. The formation of physical trees, e.g., in botanics or airway/vascular systems, is typically constrained by environmental factors such as available growth space or crowding avoidance. We, therefore, imposed some growth constraints on our synthetic trees in the form of space boundaries.

### 2.1.1 GENERATION OF SYNTHETIC TREES WITH HIDDEN MARKOV MODELING

Each Markov state represents a branching rule for bifurcations. Bifurcations in the synthetic 3-D graph-encoded trees are described using three angles (Figure 1b): (1-2) Branching angles $(\alpha_1, \alpha_2)$ measured between the direction in which each child branch is emitted, and its parent's heading direction (smallest angle being set to $\alpha_1$ to avoid inference of redundant branching rules, simply swapping $\alpha_1, \alpha_2$ values); (3) Roll or plane rotation $(\phi)$ between the plane in which the parent branch was emitted and the plane containing the two child branches.

The HMM with $R$ states is defined with the following parameters: The state-transition matrix $\boldsymbol{A} \in \mathbb{R}^{R \times R}$, where each row is drawn from an $R$-dimensional Dirichlet distribution; The vector of starting-state probabilities, $\boldsymbol{s} \in \mathbb{R}^{1 \times R}$, similarly drawn from an $R$-dimensional Dirichlet distribution. We model a Gaussian emission distribution with diagonal covariance for each Markov state, from which branching angles are drawn $\mathcal{N}(\boldsymbol{\mu}_r, \boldsymbol{\sigma}_r^2), r \in 0, 1, \dots R - 1$, where $\boldsymbol{\mu}_r = [\mu_{\alpha_1}, \mu_{\alpha_2}, \mu_\phi]_r \in \mathbb{R}^3, \boldsymbol{\sigma}_r^2 = \text{diag}([\sigma_{\alpha_1}^2, \sigma_{\alpha_2}^2, \sigma_\phi^2]_r) \in \mathbb{R}^3$. At every time-step $t$ in the growth of a new branch, the state of the branching rule $(r_t)$ is selected from a Categorical distribution with probabilities derived from the row of the state-transition matrix, $\boldsymbol{A}_{r_{t-1}} \in \mathbb{R}^{1 \times R}$ corresponding to the preceding state, $r_{t-1}$.

Two child branches are emitted using angles drawn from the state $r_t$ emission probability distribution, $\boldsymbol{y}_t = [\alpha_1, \alpha_2, \phi] \sim \mathcal{N}([\mu_{\alpha_1}, \mu_{\alpha_2}, \mu_\phi]_{r_t}, \text{diag}([\sigma_{\alpha_1}^2, \sigma_{\alpha_2}^2, \sigma_\phi^2]_{r_t}))$. The length of emitted

branches at a bifurcation are determined by the depth in the tree as detailed in Sup. B. An example synthetic binary tree is shown in Figure 1b. The full generative process is described in Algorithm 1.

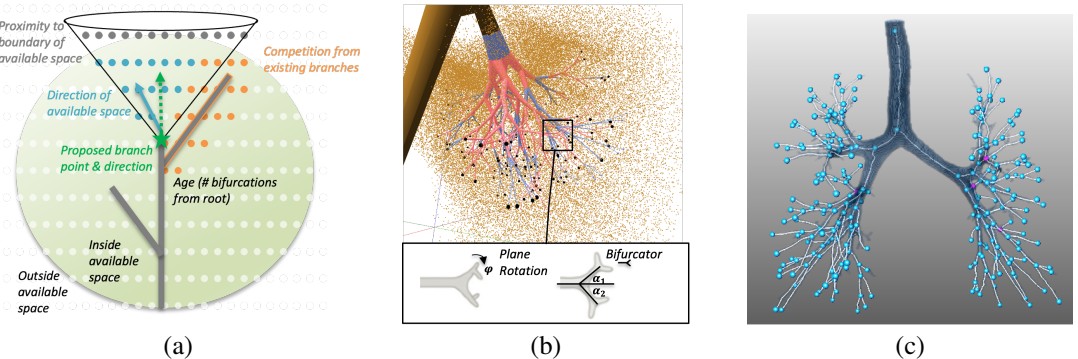

(a)            (b)            (c)

**Figure 1:** (a) **Spatial constraints for synthetic trees:** Constraints imposed on branch termination rules inside the PV cone. Green circle = total space available for growth. White dots = candidate locations for growth (e.g., voxel grid). (b) **Synthetic tree data:** Example from the Baseline dataset (Table 1) constrained by an exemplar lung region (brown point cloud), colored by HMM Rule. (c) **Human airway tree data:** Overlay of 3-D segmentation from ATM'22 and our generated centerlines. Pink nodes highlight trifurcations (here 3 trifurcations for >100 bifurcations).

### 2.1.2 SOFT SPATIAL CONSTRAINTS ON TERMINATION OF SYNTHETIC TREES

Termination of candidate branching nodes along a generated path is determined by geometric features characterizing available growth space, which are encoded in $\boldsymbol{d} \in \mathbb{R}^4 = d_1 \cdots d_4$ (as detailed below) based on measures made in a perception volume (PV) around the parent branch. We model PV space as a cone with height set to $2 \times l$, where $l$ = parent branch length and subtended angle set to $90°$. Such constraints relate to concrete use-cases such as human airway tree modeling where the proximity of growing airway branches to lung boundary, crowding between branches, and directions toward available space along branching generations play a suspected role at the embryonic stage to ensure even distribution of oxygen through the lung tissue. The following four termination features are therefore evaluated at the tip end of each candidate parent branch, in the current growing direction and within the PV space, as illustrated in Figure 1a: $d_1$ = volume fraction of available locations in PV that lie outside the available growth space (gray dots); $d_2$ = volume fraction of already 'occupied' space (orange dots). Candidate locations (e.g. voxels) closest to existing branch points are considered as occupied; $d_3$ = dot product of the parent direction and mean direction of available locations (blue dots); $d_4$ = the 'age' parameter that encodes the branch generation ($g$). The value of $g$ increases by $+1$ at bifurcation along a path from the root node (where $g = 0$). We set $d_4 = \frac{10-g}{10}$ to encourage trees with an average of 10 branching points per path.

Spatial constraints $\boldsymbol{d} = [d_1 \ldots d_4]$ are individually weighted by a scalar vector $\boldsymbol{w} = [w_1 \ldots w_4]$ to control our prior on branch termination behavior in synthetic trees and hence the overall tree characteristics such as deep or shallow, with dense or sparse spatial crowding (See exemplar values in Table 1). The values of $w_i$ are empirically determined and verified by an expert clinician to generate visually realistic trees for our purposes; examples of trees with varying $\boldsymbol{w}$ weight constraints are in Sup. C.

During HMM tree generation and inference, we encode our branching termination rules as a binary classification task. At each candidate parent branch, if $\sigma(\boldsymbol{w} \cdot \boldsymbol{d}) > 0.5$, further draws are made from the HMM; else, branching is terminated along the path.

### 2.2 HUMAN AIRWAY TREE DATASET FROM THE ATM COHORT

We tested our proposed model to infer an unknown number of HMM branching rules and state-transitions on the ATM'22 grand challenge dataset, consisting of 500 anonymized CT scans, of which 300 have reference airway segmentation (Zhang et al., 2023). Lung conditions of the scanned subjects range from healthy status to severe pulmonary disease. The segmentation quality varies

across the cohort. Since we are interested in modeling the 'native' airway tree structure (unaffected by disease), we select 50 scans from the ATM dataset with the lowest dysanapsis score (dysanapsis = small airway tree caliber relative to lung size), which is a major risk factor of incident COPD and all-cause mortality (Vameghestahbanati et al., 2023). Further details on motivations and the dysanapsis score measurements are provided in the Sup. E.1 and E.3. Details on ATM airway centerlines extraction, path encoding, and handling of trifurcations from the provided reference 3-D airway segmentations are provided in Sup. E.2.

We decompose each tree into 'paths-of-bifurcations': a set of paths from the root (first bifurcation from the trachea after the Carina) to each leaf node. The mean number of bifurcation nodes per airway tree is $239 \pm 71$, and the mean number of paths extracted per tree is $121 \pm 35$ with an average of $7.1 \pm 1.8$ branch generations per path.

## 2.3 INFERENCE

### 2.3.1 OBJECTIVE FUNCTIONS

We treat the inference of HMM parameters (requiring marginalization of latent Markov state variables) separately from the inference of global spatial constraints.

For approximate inference of the HMM model detailed in Section 2.1.1, we optimize the lower bound of Equation 1, which is the log joint distribution of the observed branching angles and latent variables in the HMM (see Table 3, Sup. B for definitions of terms):

$$
\begin{aligned}
\log p(\boldsymbol{y}, \boldsymbol{A}, \boldsymbol{s}, \boldsymbol{\mu}, \boldsymbol{r}; \mu_0, \sigma_0, \alpha_0, \beta_0, \gamma_0) &= \log p(\boldsymbol{s}; \gamma_0) + \sum_{r=1}^{R}\Bigg[\log p(\boldsymbol{A}_{r:}; \gamma_0) + \log p(\boldsymbol{\mu}_r; \mu_0, \sigma_0^2 \mathbb{I}) \\
&+ \log p(\boldsymbol{\sigma}_r^2; \alpha_0, \beta_0)\Bigg] + \sum_{i=1}^{N}\sum_{j=1}^{M}\Bigg[\log p(r_{i,j,1}|\boldsymbol{s}) + \log p(\boldsymbol{y}_{i,j,1}|\boldsymbol{\mu}_{r_{i,j,1}}, \mathrm{diag}(\boldsymbol{\sigma}_{r_{i,j,1}}^2)) \\
&+ \sum_{r=1}^{R}\sum_{t=2}^{T_{i,j}}\log p(r_{i,j,t}=r|\boldsymbol{A}_{r_{i,j,t-1}}) + \log p(\boldsymbol{y}_{i,j,t}|\boldsymbol{\mu}_{r_{i,j,t}}, \mathrm{diag}(\boldsymbol{\sigma}_{r_{i,j,t}}^2))\Bigg]
\end{aligned}
\tag{1}
$$

where indices correspond to tree $i$, path $j$, and time step on path $t = 1 \ldots T_{i,j}$; $\boldsymbol{y}_{i,j,t} = [\alpha_1, \alpha_2, \phi]_{i,j,t} \in \mathbb{R}^3$ encodes the observed branching angles; $r_{i,j,t} \in 0, 1, .., R-1$ encodes the latent Markov state categorical value among $R$ categories. We use conjugate distributions for our latent variables with uniform Dirichlet priors for the state transition and start probabilities $[p(\boldsymbol{s}; \gamma_0), p(\boldsymbol{A}_r; \gamma_0)]$ since rules are drawn from a Categorical distribution $[p(r_{i,j,1}|\boldsymbol{s}), p(r_{i,j,t}|\boldsymbol{A}_{r_{i,j,t-1}})]$, and Normal-Inverse Gamma priors $[p(\boldsymbol{\mu}_r; \mu_0, \sigma_0^2\mathbb{I}), p(\boldsymbol{\sigma}_r^2; \alpha_0, \beta_0)]$ since observed angles are drawn from Gaussian emission distributions with diagonal covariance $[p(\boldsymbol{y}_{i,j,t}|\boldsymbol{\mu}_{r_{i,j,t}}, \mathrm{diag}(\boldsymbol{\sigma}_{r_{i,j,t}}^2))]$; $(\mu_0, \sigma_0, \alpha_0, \beta_0, \gamma_0)$ are hyperparameters for the prior distributions, informed by expert clinicians and airway properties manually reported (Metzger et al., 2008).

For branch termination constraints detailed in Section 2.1.2, we infer maximum a-priori (MAP) and maximum-likelihood (MLE) estimates of the global latent variable $\boldsymbol{w} \in \mathbb{R}^4$, by optimizing the log joint distribution in Equation 2 (logistic regression),

$$
\log p(\boldsymbol{e}, \boldsymbol{d}, \boldsymbol{w}; w_0) = \overbrace{\sum_{i=1}^{4}\log p(w_i; w_0)}^{\text{log prior}} + \sum_{i=1}^{N}\sum_{j=1}^{M}\sum_{t=1}^{T}\Bigg[e_{ijt}\log \sigma(\boldsymbol{w} \cdot \boldsymbol{d}_{ijt}) + (1 - e_{ijt})\log(1 - \sigma(\boldsymbol{w} \cdot \boldsymbol{d}_{ijt}))\Bigg]
\tag{2}
$$

where indices correspond to tree $i$, path $j$ and time step on path $t = 1 \ldots T_{i,j}$; $\boldsymbol{d}_{i,j,t}$ encodes observed features in PV (Section 2.1.2) and $e_{i,j,t} \in \{0, 1\}$ encodes observed termination events

(= 0 if branching terminated, = 1 if branching continued). See Table 3, Sup. B for definitions of terms.

The log marginal likelihood (after integrating out the latent state variables) of our path-wise HMM model is differentiable and can be estimated using the HMM forward-backwards algorithm. We implemented our generative model and inference process in the scalable probabilistic programming language Pyro (Bingham et al., 2019; Obermeyer et al., 2019), with which we can apply automatic-differentiation variational inference to fit our generative model to observed paths sampled from a cohort of trees with GPU-acceleration. We also demonstrate the use of a flexible and user-friendly CPU-based inference pipeline with package *hmmlearn*[1], a *scikit-learn* (Pedregosa et al., 2011) extension for sequence data.

### 2.3.2 TRAINING SETUP

We split our datasets (synthetic and human trees) into a random 4:1 train-test split.
Across all inference experiments, we applied the following set of parameters. $w_0 = (0, 1)$; $\gamma_0 = 1$; $\alpha_0 = 1$; $\beta_0 = 0.1$; $\mu_0 = 0$; $\sigma_0^2 = 0.1$. Stochastic variational inference (SVI) is conducted on the pooled dataset of $N_{train}$ trees $\times M$ paths per tree, which varies per cases in the train set. We pre-processed angles into radians, setting $\alpha_1 < 0, \alpha_2 > 0$. We randomly sub-sample mini-batches of 200 paths and using Pyro's AutoDelta guide, we conduct gradient descent with $lr = 0.05$ and Adam optimizer for 500 epochs with early stopping.
We use the **log-likelihood** on the held-out test set to choose the best model fit out of 10 random initializations per experiment. The higher this value, the better fit of the model to unseen data.

### 2.4 SYNTHETIC TREES: BENCHMARKING EXPERIMENTS

We first verify our method on a baseline two-state HMM model (Table 1) with distinct emission distributions (Rule 0, Rule 1) for branching rules. All baseline-derived HMM models contain two states and fixed spatial constraints. All baseline-derived HMM model datasets consist of 500 trees with an average number of $103 \pm 9$ paths per tree.

**Table 1:** Baseline synthetic trees generated for a 2-state HMM tree model: Ground truth values for the transition matrix and state-based emission distributions are provided. Termination constraint values in vector $\boldsymbol{w}$ are defined in Sec 2.1.2 The number of paths per baseline tree and the baseline Kullback-Liebler divergence (KLD) between emission distributions is shown.

| Transition matrix $\boldsymbol{A}$ | Start probabilities $\boldsymbol{s}$ | Rule 0= $(\boldsymbol{\mu}_{r_0}, \boldsymbol{\sigma}_{r_0}^2)$ | Rule 1= $(\boldsymbol{\mu}_{r_1}, \boldsymbol{\sigma}_{r_1}^2)$ | Termination constraints $\boldsymbol{w}$ | $M$ = # Paths/Tree | KLD |
|---|---|---|---|---|---|---|
| $\begin{pmatrix} 0.7 & 0.3 \\ 0.3 & 0.7 \end{pmatrix}$ | $(0.5 \quad 0.5)$ | $\mathcal{N}([10, 20, 40], 5^2 \mathbb{I})$ | $\mathcal{N}([20, 40, 80], 5^2 \mathbb{I})$ | $[-0.668, -0.304, 0.304, 0.607]$ | $103 \pm 9$ | 42 |

**Table 2:** Train-time augmentation experiment parameters. All HMMs have baseline parameters (Table 1), except for variable changes indicated here.

| Experiment | Parameter | Varying Settings | Experiment | Parameter | Varying Settings |
|---|---|---|---|---|---|
| **RH-A** Rule means | $\boldsymbol{\mu}_{r_0}$ | $[12.5, 25, 50], [15, 30, 60],$ $[17.5, 35, 70], [20, 40, 80]$ | **RH-B** Rule variance | $\boldsymbol{\sigma}_{r_1}^2$ | $1^2 \mathbb{I}, 10^2 \mathbb{I}, 15^2 \mathbb{I}$ |
| **LPR-A** Low probability rules | $\boldsymbol{A}, \boldsymbol{s}$ | $\boldsymbol{A} = \begin{pmatrix} 0.2 & 0.8 \\ 0.2 & 0.8 \end{pmatrix}, \begin{pmatrix} 0.1 & 0.9 \\ 0.1 & 0.9 \end{pmatrix}, \begin{pmatrix} 0.05 & 0.95 \\ 0.05 & 0.95 \end{pmatrix}$ $\boldsymbol{s} = (0.2 \quad 0.8), (0.1 \quad 0.9), (0.05 \quad 0.95)$ | **LPR-B** Small dataset size | $\boldsymbol{A}, N_{train}$ | $\boldsymbol{A} = \begin{pmatrix} 0.1 & 0.9 \\ 0.1 & 0.9 \end{pmatrix}$ $N_{train} \in [5, 400]$ |
| **NDQ** Noisy data quality | $\mu_p$ | $[0, 0.5, 1, 1.5, 2]$ | | | |

---

[1] https://github.com/hmmlearn/hmmlearn

In a **first experiment** (Figure 2), we evaluate the impact of the presence of the **prior termination weights** $w$ used to generate synthetic trees and their inference during model fitting: MAP inference (cf. log prior term in Equation 2) with a Gaussian prior, $\mathcal{N}(0,1)$, encouraging small weights and hence early termination, MAP inference with a Laplace prior, $\text{Laplace}(0,1)$, encouraging sparsity of the influencing factors of growth and MLE inference with no prior constraints on the value of $w$.

We then run the following experiments on the **impact of synthetic data characteristics** via train-time augmentation varying one synthetic parameter at a time (Full parameter settings in Table 2):

1. **Rule heterogeneity:** manipulated via the KLD distance between emission distributions. **(RH-A)** We linearly interpolate between the emission distribution means and **(RH-B)** We change the variance of one rule emission distribution.

2. **Low probability rules: (LPR-A)** We vary the values of the start state probabilities to make state R0 rare from the beginning and change the state transition matrix, $A$, to favor state R1. **(LPR-B)** We further investigate the impact of train dataset size $N_{train}$ in this case.

3. **Noisy data quality: (NDQ)** We simulate segmentation errors on narrower branches by applying a depth-based perturbation via additive Gaussian noise on each angle drawn from the Baseline HMM emission distributions: $\alpha \sim \mathcal{N}(\mu_{r_\alpha}, \sigma_{r_\alpha}^2) + \mathcal{N}(\mu_p \times t, 1)$ where $t$ encodes depth as time of emission. We consider mild ($\mu_p \leq 1$) and extreme ($\mu_p > 1$) noise conditions.

## 2.5 EVALUATION METRICS

We report distances between emission rules using the Kullback-Liebler divergence (KLD) metric. On synthetic data, we document our best model results with the following metrics:

(1) **HMM Similarity Score** relative to ground truth parameters (Sahraeian & Yoon, 2011). This score measures whether two different HMM models are likely to be in states with similar distributions and is robust to rule permutations (0 = no similarity, 1 = perfect match).

(2) **Adjusted Rand Index** (Hubert & Arabie, 1985) for state assignment accuracy, computed using the Viterbi algorithm along each path vs ground truth assignments (0 = random chance, 1 = perfect match). We report Adjusted Rand (per path) and Adjusted Rand (consensus), where majority voting is used to finalize assigned states for branching points across paths. We also report the result of an oracle binary classifier based on observed angles and known threshold values of equal emission probabilities with selection on (i) all three angle thresholds (ii) at least two thresholds.

(3) Mean absolute error (MAE) in recovered branching rule means, standard deviations.

(4) Accuracy of the binary decision task for branch termination.

(5) We also report examples of inferred HMM parameters and termination weights.

On the human airway tree dataset, we identify the elbow in the **Bayesian Information Criterion** (BIC) score on the held-out test set for best model selection as in Mohammadi et al. (2022) and illustrated for a four-rule synthetic example in Sup D.5.

Results are summarized in Section 3 below, further details are provided in Sup. D.

# 3 RESULTS AND DISCUSSION

## 3.1 SYNTHETIC: BASELINE

From Figure 2, we see stable recovery of ground truth HMM parameters from our baseline dataset (Table 1). The estimated HMM parameters from the best performing model show strong agreement for both narrow and wider branching angles with MAE$= 0.17°$ across both $\boldsymbol{\mu}_r$ and MAE$= 0.12°$ across both $\boldsymbol{\sigma}_r$.

The Adjusted Rand Index on both per path and consensus per node bases show perfect agreement with ground truth assignments. As a comparison, we use two oracle binary classifiers which assign Rule 1 if either $(3/3)$ or $(2/3)$ of $[\alpha_1, \alpha_2, \phi] < [15, 30, 60]$ and Rule 0 otherwise. The $(3/3)$ classifier performs poorly with a score of $0.727$ and the $(2/3)$ classifier slightly under performs the HMM assignments (Figure 2b). This highlights the importance of Markovian lineage information when characterizing branching events.

Termination weights are inferred with accuracy $> 99\%$ for all inference methods (Figure 2c). MLE

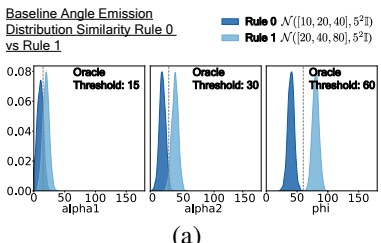 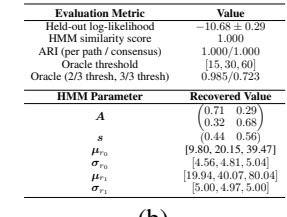 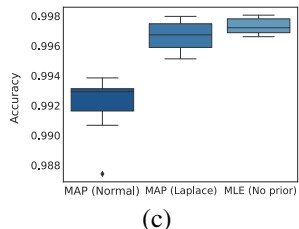

(a)                                    (b)                                    (c)

**Figure 2:** Test inference results on **baseline synthetic trees**. (a) Ground-truth angle distributions with oracle threshold. (b) Baseline evaluation metrics (Sec. 2.5), and recovered HMM parameters from best performing model. (c) Test accuracy of binary branch termination decision when varying the prior distribution of $w$.

outperforms and, therefore, is sufficient for inference of $w$. Given the high performance on $w$ inference, this could be used for characterizing trees for downstream applications, such as clustering trees based on their termination drivers in future work.

Overall, the high fidelity of inferred HMM parameters indicate that performance is not degraded by treating $w$ independently at inference, and therefore, we only infer HMM parameters in subsequent experiments.

### 3.2 Synthetic: Rule Heterogeneity (RH)

When varying the emission distribution means (Exp. RH-A, Figure 3), we see an increase in inference quality as rule heterogeneity (measured with KLD on emission distributions) increases, which is expected. We see a sharp jump in the best model HMM similarity score at KLD=3 and a steady increase across Adjusted Rand Index as KLD increases. For KLD in range $\in [3, 10]$ we see greater reliability for inferring HMM parameters versus state assignments; for KLD=10 we recover rule means with an MAE= $0.38°$ and Adjusted Rand (consensus) $= 0.962$; for KLD=3, MAE= $0.05°$ vs Adjusted Rand (consensus) $= 0.603$. The variance in held-out log-likelihood is reasonably large, showing the importance of multiple runs for convergence in cases of small heterogeneity.

When increasing the variance of one of the emission distributions (Exp. RH-B), our three evaluation metrics show distinct sensitivity (Figure 3). The held-out log-likelihood improves as variance increases, with tight distribution across runs indicating stable performance; the best model HMM similarity score starts improving and then worsens slightly (remaining strong throughout); the Adjusted Rand index is relatively strong and stable. The performance gap to the oracle $(2/3)$ classifier increases with Rule 1 variance. In the case of variance= $15^2\mathbb{I}$, HMM inference still recovers rule means with MAE= $0.61°$ and rule standard deviations with MAE= $0.96°$.

### 3.3 Synthetic: Low probability rules (LPR)

From Figure 4, we see a slight reduction in HMM similarity score as $p(r_0)$ decreases, but with still good agreements on recovered HMM parameters even with $p(r_0) = 0.05$, with overall MAE for rule means $= 0.31°$ and near perfect recovery of state assignments across experiments. Held-out log-likelihood increases with $p(r_0)$, again with small variance. When decreasing the number of training trees from 400 to 5 with $p(r_0) = 0.1$, we report the distribution in HMM similarity score across runs to highlight very stable performance of HMM inference quality even on small size training sets.

### 3.4 Synthetic: Noisy data quality (NDQ)

From Figure 5 we observe that the HMM similarity score begins to deteriorate as the noise perturbation mean increases above 0.5. For $\mu_p = 0.5$ (reasonable noise), we achieve an MAE $= 0.79°$ for the rule angle means and $0.31°$ for rule angle standard deviations (HMM score =0.995); For $\mu_p = 2$ (extreme noise, Figure 5a), we achieve an MAE $= 4.2°$ for the rule means and $2.2°$ for rule standard deviations. On the other hand, we see some improvement in held-out log-likelihood as estimated HMM parameters move away from the ground-truth to incorporate the added noise distributions. The Adjusted Rand index is remarkably stable across noise levels. State assignment performance

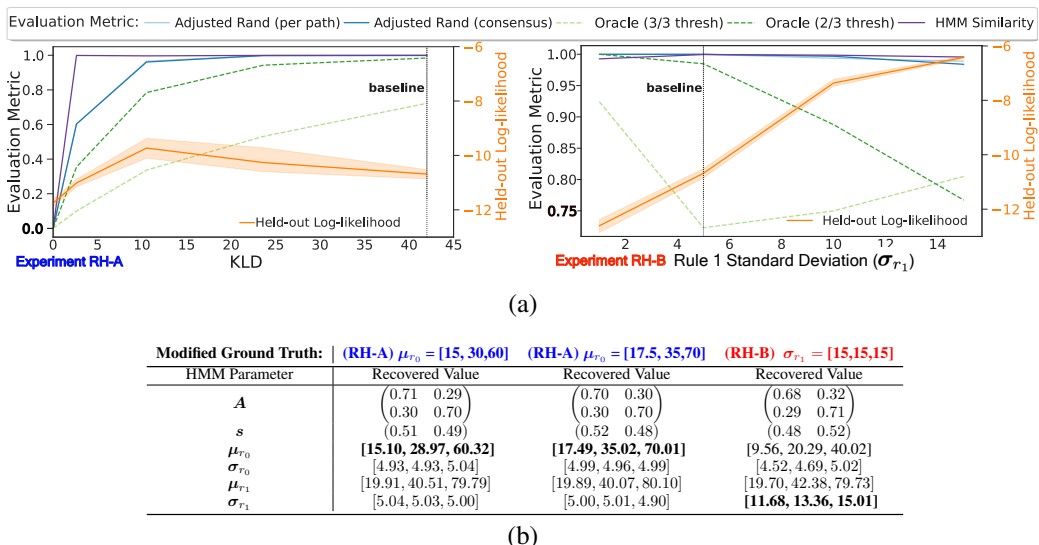

(a)

| Modified Ground Truth: | (RH-A) $\mu_{r_0} = [15, 30, 60]$ | (RH-A) $\mu_{r_0} = [17.5, 35, 70]$ | (RH-B) $\sigma_{r_1} = [15,15,15]$ |
|---|---|---|---|
| HMM Parameter | Recovered Value | Recovered Value | Recovered Value |
| $A$ | $\begin{pmatrix} 0.71 & 0.29 \\ 0.30 & 0.70 \end{pmatrix}$ | $\begin{pmatrix} 0.70 & 0.30 \\ 0.30 & 0.70 \end{pmatrix}$ | $\begin{pmatrix} 0.68 & 0.32 \\ 0.29 & 0.71 \end{pmatrix}$ |
| $s$ | $(0.51 \quad 0.49)$ | $(0.52 \quad 0.48)$ | $(0.48 \quad 0.52)$ |
| $\mu_{r_0}$ | $[15.10, 28.97, 60.32]$ | $[17.49, 35.02, 70.01]$ | $[9.56, 20.29, 40.02]$ |
| $\sigma_{r_0}$ | $[4.93, 4.93, 5.04]$ | $[4.99, 4.96, 4.99]$ | $[4.52, 4.69, 5.02]$ |
| $\mu_{r_1}$ | $[19.91, 40.51, 79.79]$ | $[19.89, 40.07, 80.10]$ | $[19.70, 42.38, 79.73]$ |
| $\sigma_{r_1}$ | $[5.04, 5.03, 5.00]$ | $[5.00, 5.01, 4.90]$ | $[11.68, 13.36, 15.01]$ |

(b)

**Figure 3:** Experiments **RH-A**, **RH-B** with changing rule heterogeneity. (a) Evaluation metrics (Sec. 2.5). (Experiment RH-A) Performance versus rule heterogeneity measured with KLD when changing Rule 0 angle emission distribution mean (baseline ground truth Rule 1 $= \mathcal{N}([20, 40, 80], 5^2\mathbb{I})$). (Experiment RH-B) Performance versus modified Rule 1 standard deviation when changing angle emission distribution variance on Rule 1 (baseline ground truth Rule 0 variance $= 5^2\mathbb{I}$). Dashed black line = baseline configuration. (b) Recovered HMM parameters from best model three experiments: (RH-A) Rule 0 $= \mathcal{N}([15, 30, 60], 5^2\mathbb{I})$ (KLD=10.5); (RH-A) Rule 0 $= \mathcal{N}([17.5, 35, 70], 5^2\mathbb{I})$ (KLD=2.62); (RH-B) Rule 1 variance $= 15^2\mathbb{I}$.

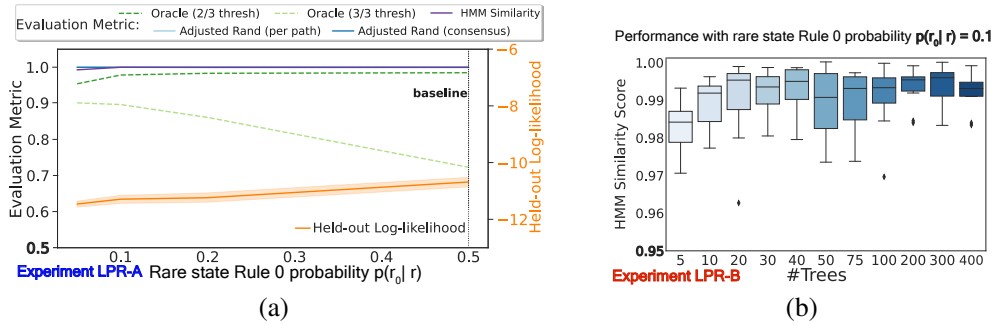

**Figure 4:** Experiments **LPR-A, LPR-B** with rare state Rule 0 ($r_0$). (a) (LPR-A) Evaluation metrics (Sec. 2.5) versus $p(r_0|r)$. Oracle threshold $[15, 30, 60]$ (baseline emission distributions). Dashed black line = baseline configuration. (b) (LPR-B) HMM similarity score versus $N_{train}$ = number of trees in the training set.

remains strong for the HMM predictions, with HMM outperforming at $\mu_p = 2$ (Figure 5). These results provide some strong confidence in the capacity of our modeling approach to handle real-world data with noisy observations.

## 3.5 HUMAN AIRWAY TREE DATASET FROM THE ATM COHORT

We varied the inferred number of branching rules (Markov states) from 2 to 19 to fit our model on the ATM dataset and used the BIC score on the held-out test set for best model selection (Figure 6). The best model fit is achieved for six branching rules based on average BIC score. The relatively low number of inferred Markov states and the sparsity of the transition matrix supports the existence of specific branching modes and transition rules in human airways. Such structuring was seen in mouse models (Metzger et al., 2008), with domain branching transition rules characterized as intermediate plane rotation or iterations between planar bifurcation and orthogonal bifurcation modes for exam-

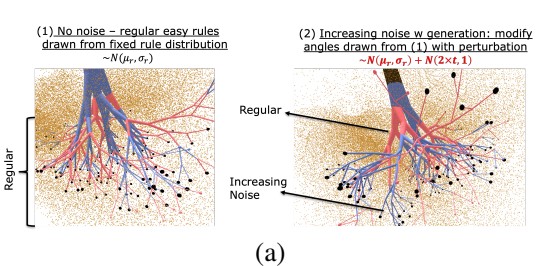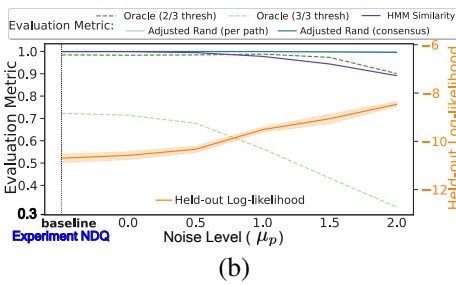

**Figure 5:** Experiment **NDQ**: (a) Simulation of a synthetic tree with baseline parameters (i.e. no noise perturbation) vs a tree generated with $\mu_p = 2$. (b) Evaluation metrics (Sec. 2.5) versus noise level added controlled by $\mu_p$ from 0 to 2) sampled from baseline ground truth values. Dashed black line = baseline configuration.

ple. Quantitative evidence of high-level structures in airway trees can support further hypothesis formulations on the biological process underpinning the generation of these structures.

Inferred HMM rule standard deviations are higher than in our 'synthetic' model (in particular for plane rotation angle $\phi$), which might be due to our noisy observation conditions (Section 3.4). Characterization of the geometry of branched structures such as airways is crucial, as small changes in geometry can result in large changes in airflow (Mauroy et al., 2004). Previous attempts to model such systems are based on complex rule systems to create an approximate fractal structure (Abbasi & Boozarjomehry, 2021; Kitaoka et al., 1999) and rely on average lung properties derived from $\approx 5$ specimens (Mauroy et al., 2004; Weibel et al., 1963) to assess model fidelity. Our probabilistic approach learns branching rules directly from data, thus leveraging large imaging cohorts. This is a crucial stepping stone towards fully data-driven generative modeling of human airway and vascular structures.

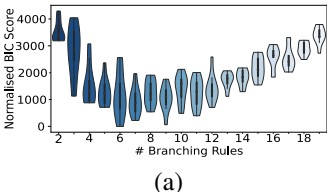

| HMM Param. | Recovered Value | HMM Param. | Recovered Value | | | | | |
|---|---|---|---|---|---|---|---|---|
| $\boldsymbol{\mu}_{r_0}, \boldsymbol{\sigma}_{r_0}$ | $[30.1, 47.3, 49.5], [9.5, 8.5, 21.3]$ | | **0.37** | 0.09 | 0.3 | 0.04 | 0.05 | 0.14 |
| $\boldsymbol{\mu}_{r_1}, \boldsymbol{\sigma}_{r_1}$ | $[33.8, 56.6, 57.1], [14.2, 13.4, 21.4]$ | $\boldsymbol{A}$ | **0.56** | 0.12 | 0.2 | 0 | 0.11 | 0 |
| $\boldsymbol{\mu}_{r_2}, \boldsymbol{\sigma}_{r_2}$ | $[18.6, 30.9, 53.9], [6.3, 7.0, 23.1]$ | | 0.01 | 0.01 | **0.55** | 0.06 | 0.26 | 0.11 |
| $\boldsymbol{\mu}_{r_3}, \boldsymbol{\sigma}_{r_3}$ | $[19.0, 47.8, 24.6], [8.0, 15.8, 14.4]$ | | 0.23 | 0 | 0 | 0.08 | 0.19 | **0.49** |
| $\boldsymbol{\mu}_{r_4}, \boldsymbol{\sigma}_{r_4}$ | $[18.4, 64.6, 25.3], [10.2, 22.8, 13.8]$ | | 0.14 | **0.45** | 0 | 0 | 0.21 | 0.19 |
| | | | **0.34** | 0 | 0.09 | **0.41** | 0 | 0.16 |
| $\boldsymbol{\mu}_{r_5}, \boldsymbol{\sigma}_{r_5}$ | $[24.7, 44.2, 79.4], [8.5, 11.1, 6.5]$ | $\boldsymbol{s}$ | (0.00 | 0.31 | 0.26 | 0.19 | 0.09 | 0.15) |

(a)          (b)

**Figure 6:** Model fit on the ATM Airway Tree Dataset. (a) Normalized BIC score versus number of inferred Markov states (# rules) on held-out data across 10 runs. BIC scores are normalized such that the smallest value = 0. Optimum at # rules = 6. (c) Recovered HMM parameters for # rules = 6. Bold = $A_{ij} > \frac{1}{3}$.

## 4 CONCLUSIONS

In this work, we proposed a novel HMM tree-based inference optimized on **'paths-of-bifurcations'** observed on binary tree structures. We optimized our model fitting formulation for the existence of hidden 'procedural modeling' rules driving bifurcation patterns, and for handling noisy observations. We first demonstrated on synthetic trees the power of our proposed inference to fit complex tree structures, handling rare state rules, large variances in state variables, and noisy tree observations. We also documented the capacity of our proposed model to infer additional hidden spatial growth constraints with low impact on inference quality and high potential for downstream future tree classification tasks. A use-case was then presented for fitting human airway trees segmented on CT scans, enabling the proposition of a number of existing branching rules based on quantitative model-fitting quality criteria. This work has potential impacts on generating truthful digital twins of human airway trees as well as further phenotyping anatomical tree patterns into sub-types with distinct clinical associations.

REPRODUCIBILITY STATEMENT

Source code required to generate synthetic trees, and reproduce HMM inference results are provided at `https://anonymous.4open.science/r/paths-of-bifurcations-hmm-3BF4`. We intend to make a public release of the code repository. Once this work is published, we will also share our augmented annotations (branching point locations and clean centerlines) on the ATM Challenge dataset with the community.

Training hyperparameters and full generative model description are provided in Sup. B. A complete description of data processing steps on the ATM dataset is provided in Sup E.2.

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

## A    ADDITIONAL RELATED WORKS

Hidden Markov Models provide a probabilistic approach to the analysis of tree structures, independent of the number of nodes in a tree, however, tree structures have also been described geometrically in Billera-Holmes-Vogtmann (BHV) tree-space (Billera et al., 2001).

Branching structures with fixed leaf sets can be compared using the BHV geodesic distance between them, and has been used for clustering phylogenetic trees for outlier detection (Zairis et al., 2016) and analysis of human airway tree structures (Wysoczanski et al., 2021; Feragen et al., 2013). Extension to partial leaf sets (i.e., trees in the dataset do not have the same terminal nodes) is not straightforward. Grindstaff & Owen (2019) propose a combinatorial algorithm that defines a search space of possible super-trees in such a case.

## B    GENERATIVE MODEL DESCRIPTION

The generative process in Algorithm 1 is implemented in L-py (Boudon et al., 2012). Synthetic trees are grown in an exemplar right lung volume from the ATM'22 grand challenge dataset (Zhang et al., 2023).

The root of all synthetic trees is positioned at the point at which the 3-D airway segmentation enters the right lung volume. At each iteration of the generative process (detailed below), two child branches are emitted using the bifurcation angles drawn from the HMM. The length (in voxels) of the emitted branches at each bifurcation is computed as the mean length of branches in the 3-D airway segmentation at the same tree-depth (number of branching generations from the root).
Each synthetic tree dataset consists of $i = 1 \dots N$ trees, each split into $j = 1 \dots M$ paths from root to each leaf node, of length $T_{i,j}$. For each time-step along each path, we record the branching angles $\boldsymbol{y}_{i,j,t} \in \mathbb{R}^3 = [\alpha_1, \alpha_2, \phi]$. For each time step, a vector of spatial constraints $\boldsymbol{d}_{i,j,t} \in \mathbb{R}^4$ is measured and $e_{i,j,t} = 0$ if the path terminates and 1 if branching continues.

Latent variables in our model are: $\boldsymbol{w}$=global termination weights (cf. Sup. C below), $\boldsymbol{\mu}$=emission probabilities for each branching rule, $\boldsymbol{A}$=state-transition probabilities between branching rules, $\boldsymbol{s}$=start probabilities for branching rules.

Equation 1 describes the joint distribution of the generative model's HMM parameters (full description of terms in Table 3). For the spatial constraints, the joint distribution is provided in Equation 2 (full description of terms in Table 3 and Sup. C). Hyperparameters: $\gamma_0 = 1$; $\alpha_0 = 1$; $\beta_0 = 0.1$; $\mu_0 = 0$; $\sigma_0^2 = 0.1$. For $\boldsymbol{w}$ inference, $w_0 = (0, 1)$ for Gaussian prior, $w_0 = (0, 1)$ for Laplace Prior.

**Table 3:** Description of mathematical terms in Equations 1, and 2 (Section 2.3.1). Terms are linked to lines in Algorithm 1 (Generative process for synthetic trees).

| Term in Equation 1. | Form | Description (Line number in Algorithm 1) |
|---|---|---|
| $p(\boldsymbol{s}; \gamma_0)$ | Dirichlet distribution (R dimensional) | Conjugate prior on HMM start probabilities. Hyperparameter $\gamma_0^\dagger$. (**Line #:6**). |
| $p(\boldsymbol{A}_{r:}; \gamma_0)$ | Dirichlet distribution (R dimensional) | Conjugate prior on each row of the transition matrix, $\boldsymbol{A}$, for HMM with $R$ branching rules. Hyperparameter $\gamma_0^\dagger$. (**Line #: 7-8**). |
| $p(\boldsymbol{\mu}_r; \mu_0, \sigma_0^2\mathbb{I})$ | Multivariate Normal (3-D) distribution with diagonal covariance | Conjugate prior on angle means per HMM rule. Hyperparameters $\mu_0, \sigma_0^\dagger$. (**Line #: 9-10**). |
| $p(\boldsymbol{\sigma}_r^2; \alpha_0, \beta_0)$ | Inverse Gamma distribution | Conjugate prior on angle variance per HMM rule. Hyperparameters $\alpha_0, \beta_0^\dagger$. (**Line #: 9,11**). |
| $p(r_{i,j,1}\|\boldsymbol{s})$ | Categorical distribution | Likelihood of the first rule on path (for tree $i$, path $j$) from start probability vector (**Line #: 16**). |
| $p(r_{i,j,t}\|\boldsymbol{A}_{i,j,t-1})$ | Categorical distribution | Likelihood of next observed HMM rule (for tree $i$, path $j$, timestep $t$) from the row of the transition matrix corresponding to the previous rule (**Line #: 21**). |
| $p(\boldsymbol{y}_{i,j,t}\|\boldsymbol{\mu}_{r_{i,j,t}}, \text{diag}(\boldsymbol{\sigma}_{r_{i,j,t}}^2))$ | Multivariate Normal (3-D) distribution with diagonal covariance | Likelihood of the observed branching angles $\boldsymbol{y}_{i,j,t} = [\alpha_1, \alpha_2, \phi]$ (for tree $i$, path $j$, timestep $t$) from the emission distribution corresponding to the current HMM rule (**Line #: 16,22**). |
| **Term in Equation 2** | **Form** | **Description (Line number in Algorithm 1)** |
| $p(w_i; w_0)$ | Normal or Laplace distribution | Prior on termination weights $\boldsymbol{w}$. Hyperparameter $w_0^\dagger$. (**Line #: 4**). |
| $\sigma(\boldsymbol{w} \cdot \boldsymbol{d}_{i,j,t})$ | Probability [0,1] | Termination probability (for tree $i$, path $j$, timestep $t$) as a weighted sum of observed termination features $\boldsymbol{d}_{i,j,t}$ (see Section 2.1.2) passed through a sigmoid activation function (**Line #: 19-20**). |

$^\dagger$*See Section 2.3.2 (Training setup) for hyperparameter settings*

---

**Algorithm 1** Generative process for synthetic trees

---

1: $LVs : \{\boldsymbol{\mu}, \boldsymbol{\sigma}, \boldsymbol{A}, \boldsymbol{s}, \boldsymbol{w}\}$
2: Dataset of M paths per tree, for N trees:
3: $\left\{\boldsymbol{y}_t \in \mathbb{R}^3 = [\alpha_1, \alpha_2, \phi], \boldsymbol{d}_t \in \mathbb{R}^4\right\}_{ij}$ for $i = 1...N, j = 1...M, t = 1...T_{ij}$
4: Draw $w_i \sim \mathcal{N}(0, 1)$ {Termination probabilities $i \in \mathbb{R}^4$}
5: Initialize Markov model:
6: Draw $\boldsymbol{s} \sim Dir_R(\gamma_0)$ {Initial rule probabilities, $\gamma_0 \in \mathbb{R}^R = [1, 1..1]$}
7: **for** each rule $r \in 1..R$ **do**
8:     Draw $\boldsymbol{A}_{r:} \sim Dir_R(\gamma_0)$
9:     **for** each angle $[\alpha_1, \alpha_2, \phi]$ **do**
10:         Draw $\mu_{\alpha,r} \sim \mathcal{N}(\mu_0, \sigma_0^2)$
11:         Draw $\sigma_{\alpha,r}^2 \sim InverseGamma(\alpha_0, \beta_0)$
12:     **end for**
13: **end for**
14: **for** each participant tree $i \in 1..N$ **do**
15:     **for** each path in tree $j \in 1..M$ **do**
16:         Draw $r_{i,j,1} \sim Categorical(\boldsymbol{s})$
17:         Draw $\boldsymbol{y}_{i,j,1} \sim \mathcal{N}(\boldsymbol{\mu}_{r_{i,j,1}}, \text{diag}(\boldsymbol{\sigma}_{r_{i,j,1}}^2))$
18:         **for** each time-step $t \in 2..T_{ij}$ **do**
19:            $p(e_{ijt} = 1) = \sigma(w \cdot d_{ijt})$
20:            **if** $p(e_{ijt}) > 0.5$ **then**
21:                Draw $r_{i,j,t} \sim Categorical(\boldsymbol{A}_{r_{ij,t-1}})$
22:                Draw $\boldsymbol{y}_{i,j,t} \sim \mathcal{N}(\boldsymbol{\mu}_{r_{i,j,t}}, \text{diag}(\boldsymbol{\sigma}_{r_{i,j,t}}^2))$
23:            **end if**
24:         **end for**
25:     **end for**
26: **end for**

---

## C VARIATION OF SOFT SPATIAL CONSTRAINTS

Spatial constraints $\boldsymbol{d} = [d_1 \ldots d_4]$ are are individually weighted by scalar values in vector $\boldsymbol{w} = [w_1 \ldots w_4]$ to control our prior on branch termination behavior in synthetic trees and hence the overall tree characteristics such as deep or shallow, with dense or sparse spatial crowding. Branching terminates if $\sigma(\boldsymbol{w} \cdot \boldsymbol{d}) < 0.5$; to encourage termination, we drive $\boldsymbol{w} \cdot \boldsymbol{d} < 0$; to encourage further tree growth, we drive $\boldsymbol{w} \cdot \boldsymbol{d} > 0$. Parameters $\boldsymbol{w}$ can vary based on the shape and size of the constraining volume, the location of the root-node, and the desired density and size of synthetic trees.

Examining each component $w_i, d_i$ in turn:

- $w_1$: As $d_1$(the volume fraction of voxels in the PV that lie outside the available space) increases, we want to encourage termination (avoid branches escaping the space). Therefore $w_1 < 0$. If the absolute value of $w_1$ is too small, branches will escape the available space, and if it is too large, trees will be shallower and terminate closer to the center of the available space.

- $w_2$: As $d_2$ (the volume fraction of voxels in the PV that are occupied by existing branches) increases, we want to encourage termination (avoid spatial crowding/branch overlap). Therefore, $w_2 < 0$ and reducing the absolute value of $w_2$ leads to higher branch density.

- $w_3$: As $d_3$ (the dot product of the mean direction of available space vs. current direction) increases, we want to encourage growth (maximize space utilization). Therefore $w_3 > 0$. Reducing the absolute value of $w_3$ increases branch density.

- $w_4$: As $d_4$ encodes the age of a branch, and is positive until a pivot generation ($g = 10$ in our trees), and negative for $g > 10$, we have $w_4 > 0$ to encourage growth up-to generation 10. Increasing the absolute value of $w_4$ can increase the forcing function for growth at small generations (in spite of unfavorable spatial conditions $d_1 \ldots d_3$). The pivot generation can also be varied to modify the tree size.

Appropriate configurations for parameters $\boldsymbol{w}$ are non-unique. Distinct parameter regimes converge to similar tree morphology due to the related nature of volume fractions in the PV. We demonstrate the impact of varying $w_1, w_2, w_3, w_4$ in Figure 7. Varying $w_3$ has a smaller impact, as $d_3$ is closely related to the values of $d_1, d_2$.

## D ADDITIONAL EXPERIMENTAL RESULTS

In the following sections, we support the discussion in Section 3 with additional experimental results for synthetic experiments.

### D.1 SYNTHETIC: BASELINE

For baseline synthetic trees (Table 1, Section 3.1 of main paper), we demonstrate high performance on inference of termination weights, with $> 99\%$ test accuracy on binary termination decisions when varying the prior distribution of $\boldsymbol{w}$ (Figure 2c in Section 3.1).
Table 4 below reports the values of recovered termination weights across prior distributions on $\boldsymbol{w}$. MLE outperforms for $\boldsymbol{w}$ inference with MAE = 0.01. Gaussian prior achieves MAE= 0.02, Laplace prior underperforms with MAE=0.34.

**Table 4:** Test inference results on **baseline synthetic trees**: Inferred termination weights, $\boldsymbol{w}$, when varying prior distribution on $\boldsymbol{w}$. Experimental details are in Section 2.4.

| Inference experiment | Recovered termination weights | | | |
| --- | --- | --- | --- | --- |
| | $w_1$ | $w_2$ | $w_3$ | $w_4$ |
| MAP (Laplace Prior) | -0.673 | -0.304 | 0.283 | 0.612 |
| MAP (Gaussian Prior) | -0.684 | -0.300 | 0.247 | 0.617 |
| MLE (No Prior) | -0.672 | -0.305 | 0.287 | 0.611 |
| **Ground truth $w$** | **-0.668** | **-0.304** | **0.304** | **0.607** |

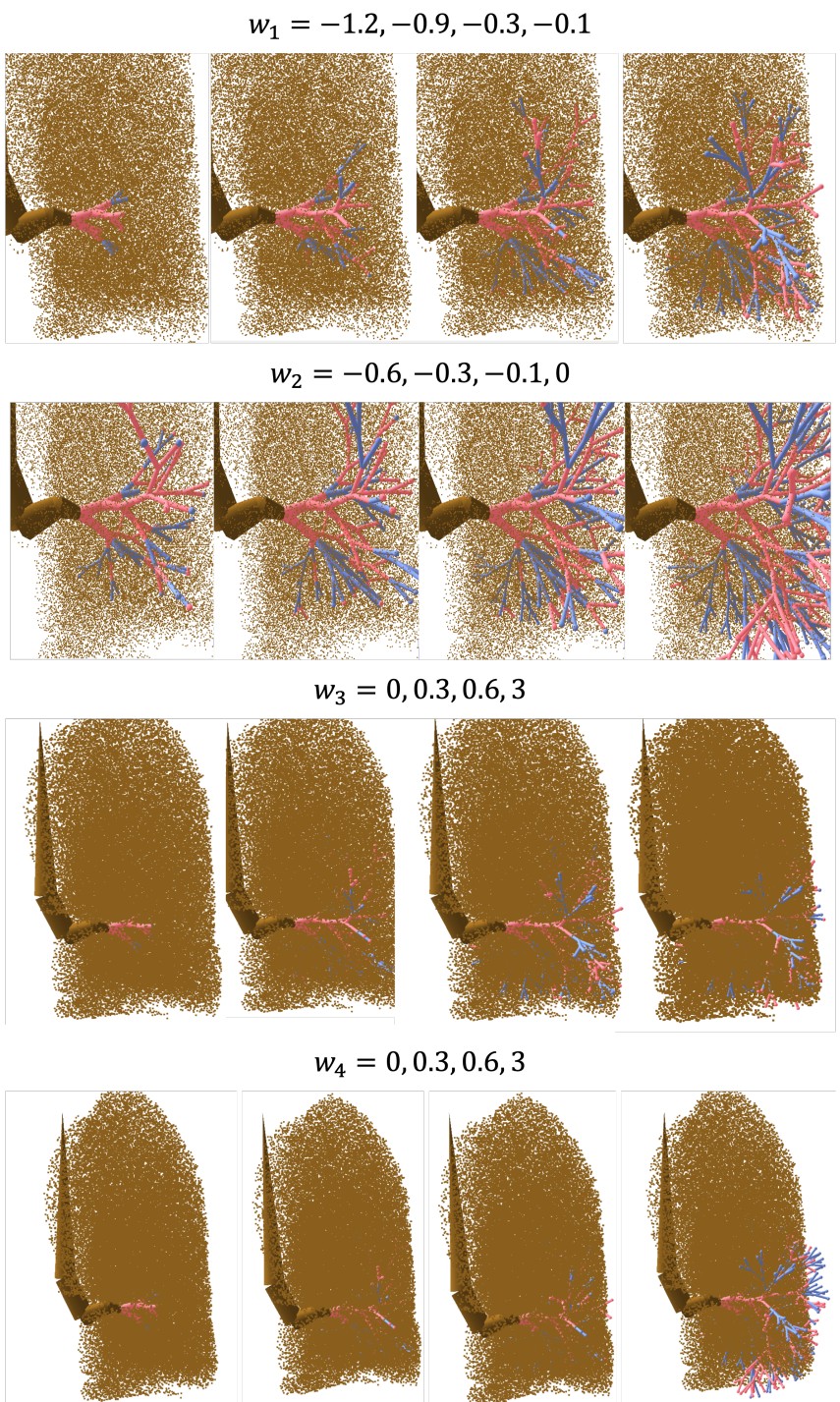

**Figure 7:** Varying termination constraints $w$ in an exemplar right lung (brown point cloud). Synthetic binary trees are generated from baseline HMM parameters in Table 1. All images are adapted from a base case $w = [-0.6, -0.3, 0.3, 0.6]$, with one $w_i$ varying per row. Zoom factors are varied to highlight boundary vs internal behavior. As $w_1$ becomes less negative, branching approaches the boundary of the available space. As $w_2$ becomes less negative, branching structure becomes more dense. As $w_3$ becomes more positive, tree growth is encouraged. However, as the branches approach the border of the space, the available space is 'behind' the branches and they terminate. As $w_4$ becomes more positive, the age-related growth driver overwhelms unfavorable spatial conditions leading to branches overlapping and escaping the available space. There is a trade-off between the $w_i$ to influence tree morphology.

## D.2 SYNTHETIC: RULE HETEROGENEITY (RH)

We report additional results for Rule Heterogeneity synthetic experiments (Section 3.2 of main paper).

For Rule Heterogeneity (RH-A) experiments changing angle emission distribution means, Figure 8 details tested angle emission distributions. For Rule Heterogeneity (RH-B) experiments changing angle emission distribution variance on Rule 1 (baseline ground truth Rule 0 variance = $5^2\mathbb{I}$), Figure 9 shows tested angle emission distributions.

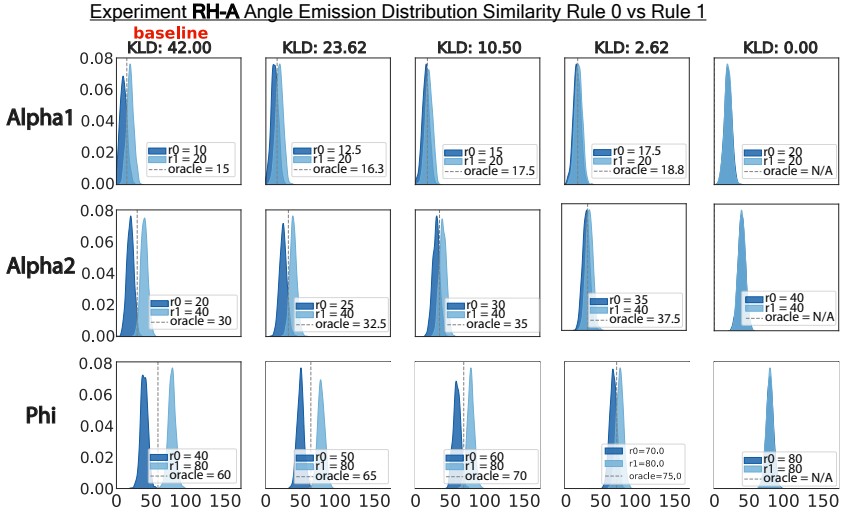

**Figure 8:** Rule Heterogeneity (RH-A) experiments. Tested angle emission distributions ordered by KL divergence (KLD) between emission means with oracle threshold. Rule 0 mean is linearly interpolated between Baseline Rule 0, $\mathcal{N}([10, 20, 40], 5^2\mathbb{I})$, and Baseline Rule 1 values, $\mathcal{N}([20, 40, 80], 5^2\mathbb{I})$. Rule 1 remains constant at baseline values, $\mathcal{N}([20, 40, 80], 5^2\mathbb{I})$. Baseline distributions are indicated in red.

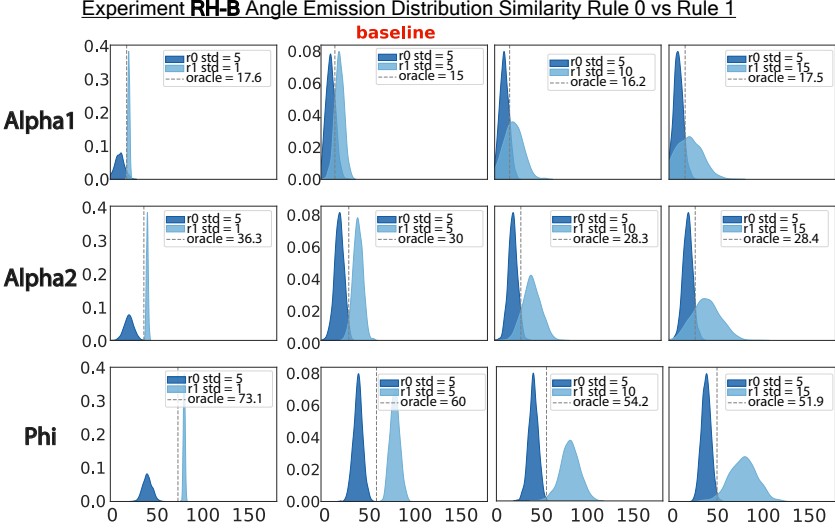

**Figure 9:** Rule Heterogeneity (RH-B) experiments. Tested angle emission distributions ordered by Rule 1 standard deviation with oracle threshold. Baseline ground truth Rule 0 variance = $5^2\mathbb{I}$, baseline ground truth Rule 0 mean =[10, 20, 40] and Rule 1 mean = [20, 40, 80]. Baseline distributions are indicated in red.

| Observed rule proportions | | | | Modified Ground Truth: (LPR-A) $p(r_0)$ = 0.05 | |
| --- | --- | --- | --- | --- | --- |
| | Start probabilities | Fraction $r_0$ (per node per tree) | Fraction $r_0$ (per node per path per tree) | HMM Parameter | Recovered Value |
| | | | | $A$ | $\begin{pmatrix} \mathbf{0.14} & 0.86 \\ \mathbf{0.05} & 0.95 \end{pmatrix}$ |
| | $[0.5, 0.5]$ | 0.491 | 0.457 | $s$ | $(\ \mathbf{0.04}\quad 0.96\ )$ |
| | | | | $\boldsymbol{\mu}_{r_0}$ | $[9.81, 20.21, 40.24]$ |
| | $[0.2, 0.8]$ | 0.202 | 0.194 | $\boldsymbol{\sigma}_{r_0}$ | $[4.69, 5.04, 7.80]$ |
| | | | | $\boldsymbol{\mu}_{r_1}$ | $[20.02, 41.21, 80.01]$ |
| | $[0.1, 0.9]$ | 0.1 | 0.098 | $\boldsymbol{\sigma}_{r_1}$ | $[4.95, 5.02, 5.00]$ |
| (a.1) | (a.2) | | | (b) | |

**Figure 10:** Experiment **(LPR-A)**. (a.1) Illustrative example tree with $p(r_0) = 0.1$, colored by HMM rule, with observed rule proportions in (a.2). (b) Recovered HMM parameters from best performing model with $p(r_0) = 0.05$.

### D.3 SYNTHETIC: LOW PROBABILITY RULES (LPR)

We report additional results for Experiment LPR-A with rare state Rule 0 (Section 3.3 of main paper).

Figure 10 details a visualization of an LPR-A synthetic tree with $p(r_0) = 0.1$, observed rule proportions in synthetic dataset with varying $p(r_0)$ and recovered HMM parameters for $p(r_0) = 0.05$. Even with $p(r_0) = 0.05$, our overall MAE for rule means = 0.31.

### D.4 SYNTHETIC: NOISY DATA QUALITY (NDQ)

We report additional results for Noisy Data Quality (NDQ) experiment (Section 3.4 of main paper). Table 5 details recovered HMM parameters for the most extreme noise case $\mu_p = 2$. Even under these conditions, we achieve MAE=4.2° for rule means and MAE=2.2° for rule standard deviations.

**Table 5: (NDQ)** Recovered HMM parameters for $\mu_p = 2$.

| Modified Ground Truth: (NDQ) $\mu_p$ = 2 | |
| --- | --- |
| HMM Parameter | Recovered Value |
| $A$ | $\begin{pmatrix} 0.69 & 0.31 \\ 0.29 & 0.71 \end{pmatrix}$ |
| $s$ | $(0.45\quad 0.55)$ |
| $\boldsymbol{\mu}_{r_0}$ | $[7.85, 24.31, 48.14]$ |
| $\boldsymbol{\sigma}_{r_0}$ | $[4.98, 7.24, 6.81]$ |
| $\boldsymbol{\mu}_{r_1}$ | $[18.72, 41.47, 87.87]$ |
| $\boldsymbol{\sigma}_{r_1}$ | $[8.67, 8.82, 6.82]$ |

### D.5 MODEL SELECTION VIA BAYESIAN INFORMATION CRITERION (BIC)

We verify our method of model selection on synthetic data, as in Mohammadi et al. (2022). We generate a synthetic cohort of 500 trees from a 4-state HMM with parameters in Table 6. Angle distributions are selected, as in Section 2.4, to align with branching rules in mammalian airway trees (Metzger et al., 2008). Soft termination weights $w$ are empirically determined to generate visually consistent trees. The experimental inference framework is similar to in Section 3, with a 4:1 train-test split and 10 random initializations. We vary the number of inferred HMM states $R \in [2, 9]$ and report the Bayesian Information Criterion (BIC) score (Cherkassky & Ma, 2003) on the held-out test set in Figure 11. We see an optimum in BIC value at 4 Rules. The best model at $R = 4$ achieves an MAE $0.54°$ for rule means and $0.26°$ for rule standard deviations.

**Table 6:** Synthetic trees generated for a 4-state HMM tree model: Ground truth values for the transition matrix and state-based emission distributions are provided. Termination constraint values in vector $\boldsymbol{w}$ are defined in Sec 2.1.1.

| Transition Matrix, $\boldsymbol{A}$ | Rules $0\ldots3$, $(\boldsymbol{\mu}_r, \boldsymbol{\sigma}_r^2)$ | Termination constraints, $\boldsymbol{w}$ |
|---|---|---|
| $\begin{pmatrix} 0.7 & 0.1 & 0.1 & 0.1 \\ 0.1 & 0.7 & 0.1 & 0.1 \\ 0.1 & 0.1 & 0.7 & 0.1 \\ 0.1 & 0.1 & 0.1 & 0.7 \end{pmatrix}$ | $\begin{aligned} &\mathcal{N}([20,40,80], 5^2\mathbb{I}) \\ &\mathcal{N}([10,60,20], 5^2\mathbb{I}) \\ &\mathcal{N}([10,20,40], 5^2\mathbb{I}) \\ &\mathcal{N}([45,45,90], 5^2\mathbb{I}) \end{aligned}$ | $[-0.3780, -0.3780, 0.3780, 0.7559]$ |

(a)           (b)

| HMM Param. | Recovered Value | HMM Param. | Recovered Value |
|---|---|---|---|
| $\boldsymbol{\mu}_{r_0}, \boldsymbol{\sigma}_{r_0}$ | $[20.05, 39.93, 79.99], [5.11, 5.07, 5.00]$ | | $\begin{pmatrix} 0.65 & 0.12 & 0.1 & 0.12 \\ 0.12 & 0.65 & 0.1 & 0.12 \\ 0.13 & 0.14 & 0.59 & 0.14 \\ 0.13 & 0.13 & 0.1 & 0.65 \end{pmatrix}$ |
| $\boldsymbol{\mu}_{r_1}, \boldsymbol{\sigma}_{r_1}$ | $[10.07, 59.94, 20.01], [4.77, 4.90, 4.97]$ | $\boldsymbol{A}$ | |
| $\boldsymbol{\mu}_{r_2}, \boldsymbol{\sigma}_{r_2}$ | $[10.02, 20.44, 40.05], [4.58, 4.75, 5.04]$ | | |
| $\boldsymbol{\mu}_{r_3}, \boldsymbol{\sigma}_{r_3}$ | $[42.10, 47.79, 89.97], [4.05, 4.10, 5.06]$ | $\boldsymbol{s}$ | $(0.27 \quad 0.26 \quad 0.20 \quad 0.27)$ |

(c)

**Figure 11:** Test inference results on 4-state HMM synthetic trees while varying the number of inferred HMM states: (a) Illustrative example tree generated using parameters in Table 6. (b) BIC score across 10 random initializations versus number of Markov states (branching rules) $= R \in [2,9]$. BIC scores are normalized such that the smallest value $= 0$. Optimal BIC value is reached for # rules $= 4$, consistent with parameters in Table 6. (c) Recovered HMM parameters for # rules $= 4$.

# E  HUMAN AIRWAY TREE DATASET: PRE-PROCESSING PIPELINE

## E.1  BACKGROUND

A growing body of evidence suggests that developmental variants in airway tree structures are associated with adverse health outcomes later in life, including all-cause mortality and incident COPD (Smith et al., 2020; 2018; Vameghestahbanati et al., 2023). The airway tree is a complex structure exhibiting multi-scale self-similarities. Assessment of inter-individual differences, remains rudimentary to-date (e.g. mean airway caliber, presence of or absence of a segmental bronchus in the lower lobe). Development of a biologically-informed generative model of airway tree structure may facilitate the categorization of airway tree sub-types and the discovery of specific characteristics associated with symptoms and genetic basis.

## E.2  SKELETONISATION & PATH ENCODING

We pre-process the 300 airway ground truth segmentations from the ATM dataset by skeletonizing the provided airway segmentation masks to extract centerlines and identify branching points using the method detailed in Selle et al. (2002). Outside of the trachea, we prune leaf nodes with length

$< 5mm$. In a few cases, where there is an trifurcation, we randomly split the juncture into two adjacent bifurcations (Figure 12).

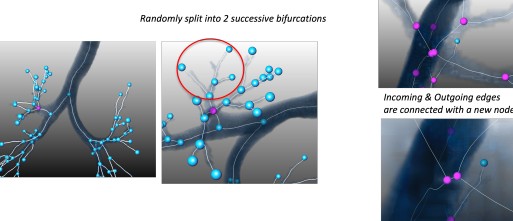

**Figure 12:** Pre-processing of trifurcations on the airway tree data (1-2 per tree on average).

For each airway segment in the pruned tree, the end-to-end vector direction is computed, and from this, the bifurcation angles $\alpha_1, \alpha_2, \phi$ are computed (Figure 13). We use the Networkx package [2] to split the trees into paths from the carina (first bifurcation of the trachea) to each leaf node. Note: paths actually start at the 2nd branching from the trachea into left and right mainstem bronchi as the 1st branching point (called the carina) lacks a parent plane rotation $\phi$. We verified the quality of all airway extracted paths with an expert clinician.

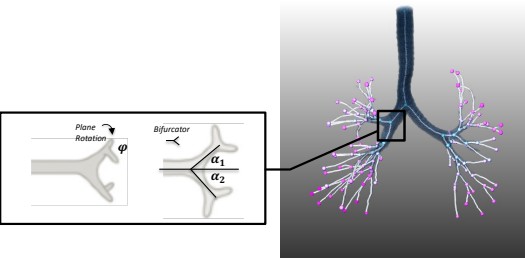

**Figure 13:** Illustration of branching angles $(\alpha_1, \alpha_2, \phi)$ extracted from airway segmentation. Figure adapted from Metzger et al. (2008).

### E.3 DYSANAPSIS SCORE COMPUTATION

The manually-measured dysanapsis score used in (Vameghestahbanati et al., 2023; Smith et al., 2020) is approximated in our work as the ratio of the geometric mean of airway lumen diameters up to the $5^{th}$ generation of segmented branches (in lieu of using anatomically labeled branches). Airway lumen diameters (in mm) are estimated at each point on the airway centerline (Selle et al., 2002). We compute the average diameter for the middle $2/3$ of each branch (to mimic locations picked by clinicians for their manual measures). 3-D lung segmentation masks are generated for the 300 CT scans using the method in (Hofmanninger et al., 2020), from which total lung volume (TLV) is computed, and lung size proxy (mm) $= \sqrt[3]{\text{TLV}}$. The distribution of dysanapsis scores on the ATM cohort is shown in Figure 14, and samples from the top and bottom 20% of dysanapsis scores measured on ATM airway trees are displayed in Figure 15. We observe some potential bias between the number of segmented branches and the top vs. bottom dysanapsis scores, as cases with more airway generations segmented seem to be more likely to return high dysanapsis scores. These cases are more challenging to generate from CT images and therefore less likely to be present in the training set.

---

[2]https://networkx.org

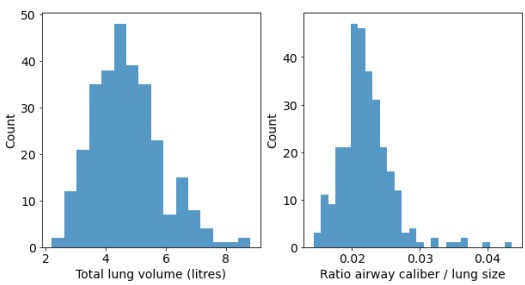

**Figure 14:** Distribution of lung volume and dysanapsis score (airway-to-lung size ratio) in the ATM dataset.

*Example Airway Trees from Top 20% of Dysnapsis Scores*

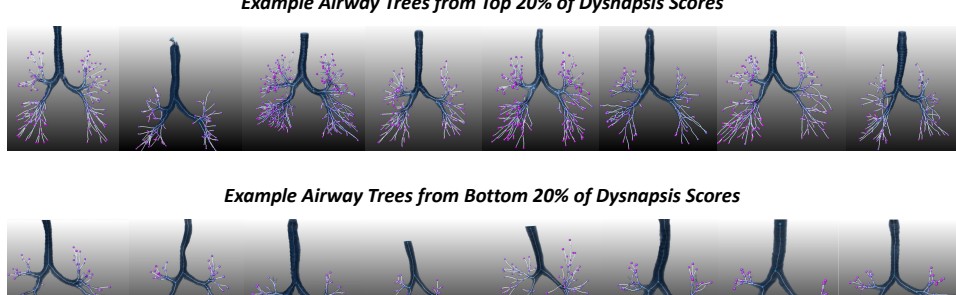

*Example Airway Trees from Bottom 20% of Dysnapsis Scores*

**Figure 15:** Illustrative examples of the large variations in branching structures in the human lung from the ATM dataset reference segmentations (Zhang et al., 2023). Airway centerlines are shown in white, with bifurcations and terminal nodes colored by airway generation from blue = 0 at the top to pink = maximum branch generation for the tree c $7.4 \pm 1.7$. The first row of airway trees are sampled from the top 20% based on dysanapsis score, the second row from the bottom 20%.

