# OpenReview forum: "Demonstrating the capacity of a Path-Based variational inference formulation for robust hidden Markov modelling of complex and noisy binary trees"
_ICLR.cc/2024/Conference — Submitted to ICLR 2024_

### Official Review · Reviewer_1unB · 2023-10-31

**Soundness:** 4 excellent
**Presentation:** 2 fair
**Contribution:** 3 good
**Rating:** 5
**Confidence:** 3

**Summary:**

The paper explores a so-called paths-of-bifurcations approach to modelling tree structures with Hidden Markov Models (HMMs). Treating each bifurcation as a node and path from root to leaf in the tree as a sequences of nodes on path to be modelled by the HMM. The approach is evaluated in its ability to infer properties of synthetic trees generated using a procedural modelling approach developed for the purpose as well as human airway trees extracted from a public medical dataset.

**Strengths:**

- Generative modelling of tree structures is an important problem and is relevant.

- Novelty appear to be reasonably clearly described, and described as contributions covering the paths-of-bifurcations approach, which is an original idea as far as I know, a procedural approach to synthetic tree generation, and the practical application to human airway trees.

- Language, grammar and structure is of high quality.

- The path-of-bifurcations concept is an interesting approach to modelling tree structures with HMM sequential modelling and appears to be novel.

- Code appears to be available, which should make the results easier to reproduce.

**Weaknesses:**

- I found the manuscript to be a relatively hard read. Methodological and implementation details are often not explained or reasoned about and there are many of them, moreover, figures and tables are too full of details. I would suggest this is a presentation problem, not all numbers and methodological details are equally important, and I feel like the authors could have made more of an effort in summarizing and selecting the important aspects. I believe this could make the work more interesting.

- Almost all tables and figures are not readable except in very enlarged versions. Could they perhaps be changed and/or resized to be readable at reasonable magnification/print size?

**Questions:**

- Could some intuition or reasoning be provided for the specific employed priors and emission distributions?

---

> ### Author Response · Authors · 2023-11-22
> **Response to review of Submission6037 by Reviewer 1unB**
>
> We thank the reviewer for their detailed comments and suggestions. We appreciate your in-depth understanding of the novelty and originality of our work, as well as the importance of generative modeling frameworks for tree structures.
>
> We greatly appreciate your feedback and provide detailed responses below, indicating modifications made to the revised manuscript. **Please note that we were granted an extension until December 1st because of a third review which was only made available to us yesterday, therefore the final manuscript may be further improved.**
>
> **[R2-1] I found the manuscript to be a relatively hard read. Methodological and implementation details are often not explained or reasoned about and there are many of them**
>
> We appreciate this feedback and aimed to address your concerns as follows:
>
> - **Methodological Explanations:**  We have added details on the HMM model and spatial constraints design to Section 1 (Introduction); and Section 2.1.2 (Soft spatial constraints on termination of synthetic trees).
> - **Methodological Reasoning:** During development, we relied heavily on domain knowledge from our expert clinician co-author, who has > 10 years of experience in airway morphology, as well as medical literature, to ensure that our generative process includes all the variables and conditioning needed to yield high-fidelity synthetic airway trees.
> - **Implementation reasoning:**
>     - **Generative model implementation:** We relied on domain-knowledge (from expert clinician and medical literature) to design a generative model suitable for human airway trees.
>     - **Validation experiments:** Benchmarking experiments showcase key challenges that exist in binary tree-structured data observed in 3-D medical images e.g., CT modalities. We handle real-case scenarios with noisy observations.
>
> - **Implementation details:** We designed our implementation to integrate existing specialized open-source libraries.
>
>
> **[R2-2] …, moreover, figures and tables are too full of details. I would suggest this is a presentation problem, not all numbers and methodological details are equally important, and I feel like the authors could have made more of an effort in summarizing and selecting the important aspects. I believe this could make the work more interesting. Almost all tables and figures are not readable except in very enlarged versions. Could they perhaps be changed and/or resized to be readable at reasonable magnification/print size?**
>
> - We appreciate this important constructive comment. We have taken extensive measures to improve data presentation and summarization throughout the paper and ensure that all tables and figures are now readable at print size, especially in Section 3 (Results and Discussion).
> -  We have added a section in Sup. D (Additional experimental results) for further details supporting the main paper.
>
> **[R2-3] Could some intuition or reasoning be provided for the specific employed priors and emission distributions?**
>
> - Our choice for the employed priors and emission distributions is based on previous literature [1,2, 3] and medical expert supervision during development to confirm fidelity of generated trees (as in **[R2-1]**).
> - We have added argumentation supporting our choice of priors in Section 1 (Introduction), Section 2.1.2 (Soft spatial constraints on termination of synthetic trees), and Section 2.3 (Inference).
>
> **References:**
>
> [1] Metzger RJ, Klein OD, Martin GR, Krasnow MA. The branching programme of mouse lung development. Nature. 2008;453(7196):745-50.
>
> [2] Abbasi Z, Boozarjomehry RB. Modeling of human conducting airways by stochastic parametric L-system. The European Physical Journal Plus. 2021;136(2):197.
>
> [3] Weibel ER, Cournand AF, Richards DW. Morphometry of the human lung. Berlin: Springer; 1963.
>
> Once again, we sincerely thank the reviewer for the detailed, constructive comments and suggestions to improve our paper.

---

> > ### Author Response · Authors · 2023-11-28
> > **Request for feedback on our response**
> >
> > Dear Reviewer,
> >
> > We greatly appreciate your suggestions which have improved our work. As the author interaction is coming to an end (our extended deadline is 1st December), we are waiting for your post-rebuttal response. If our response answers your concerns, please consider raising your scores. If not, please let us know of your further questions such that we can answer them in the remaining time.
> >
> > Kind Regards,

---

> > > ### Author Response · Authors · 2023-11-28
> > > **Request to read ongoing discussion**
> > >
> > > Dear Reviewer,
> > >
> > > We are in discussions with Reviewer xwzK. We would greatly appreciate if you could take the time to read this discussion before you submit your final scores on our paper.
> > >
> > > Once again, many thanks for your suggestions and constructive feedback on our paper.
> > >
> > > Kind Regards,

---

### Official Review · Reviewer_T8XB · 2023-10-31

**Soundness:** 3 good
**Presentation:** 2 fair
**Contribution:** 1 poor
**Rating:** 5
**Confidence:** 2

**Summary:**

The paper presentes a HMM approach to modeling binary trees. Approximate inference is performed using SVI in Pyro. The method is demonstrated on a broad array of synthetic data examples as well as a real human airways dataset.

**Strengths:**

The proposed method is fairly simple, and builds on a well known methodology.

Studies on simulated data help to illustrate the strenghts and limitations of the approach.

Simple and effective models for tree structured data seem to be important for gaining statistical insights into important biological structures.

**Weaknesses:**

There is no direct comparison with other competing approaches or baselines.

The technical novelty is limited.

The method could be more clearly described in the main paper, and variables/distributions could be more clearly defined.

**Questions:**

"Fast and scalable inference of a generative process governing such tree-structures is key for clas-
sifying trees into categories." Are generative process modeling the prevalent approach to classification, or are there other (discriminative) approaches?

"...  and split trees into ‘independent’ paths of bifurcations ..." why is independent in quotations?

Would it be possible in the introduction to set up a more clear problem formulation so that is evident which types of data problem the paper addresses? For example, it is not clear whether the trees that are modelled have a graph-based or geometric representation, such as 3d coordinat-based or anatomical mesh representation.

If I understand correctly, emission probabilities for branching angles and roll are Gaussian. Is anything done to ensure that the first angle "is the smallest angle by definition" and that angles are within a suitable range?

Would it be possible to provide some more details (possibly in the supplement) regarding the definitions of each term in the joint distribution (Eq. 1), such as a list with clear definitions of all variables and distributions.

Which variational distributions ('guide' in Pyro terminology) are used, and how important/sensitive is the choice?

Have any other methods been used to model e.g. the human airways data? How does the proposed method perform in comparison?

What would be the primary application of the proposed method?

Minor issues:
There are a few typos etc. Examples from abstract and introduction:
"...prevalent multiple across fields..." ?
Closing parenthesis doubled "))" in a few citations.
"... and infer states..." should be 'and infers states'
"... human airway trees segmented on CT images." unclear

---

> ### Author Response · Authors · 2023-11-22
> **Response to review of Submission6037 by Reviewer T8XB [Part 1/3]**
>
> We thank the reviewer for their detailed comments and suggestions.
>
> - Thank you for recognizing the wide array of synthetic data examples used to illustrate the strengths and limitations of our proposed modeling approach, alongside our human airway dataset application.
> - We also appreciate your acknowledgement of the need for simple and effective models for tree structured data to gain more biological insights.
>
> We have prepared a detailed response below and have extensively revised the manuscript to address your constructive comments and suggestions. **Please note that we were granted an extension until December 1st because of a third review which was only made available to us yesterday, therefore, the final manuscript may be further improved.**
>
> **[R1-1] There is no direct comparison with other competing approaches or baselines.**
> - The closest baseline for tree-HMMs modeling is cited in our paper as (Mohammadi et al., 2022) [1]. We tested this approach during our initial development but it did not work properly on our trees and the size of our dataset for the following reasons: CPU-implementation, design choices too domain specific (branching rules controlled by a genetically encoded subroutine).
> - We aim to simplify a specific modeling problem and integrate its implementation with open-source specialized libraries. Showing that this simplified modeling formulation provides correct and robust fits is our main goal.
> - In this context, we used synthetic benchmarking, a validation technique applied in studies such as that of Mohammadi et al. (2022) [1].
> - We develop this method in an academic setting, with limited computational resources, which limits our capacity to test complex models.
>
> **[R1-2] The technical novelty is limited.**
>
> We appreciate the reviewer's assessment regarding the novelty of our proposed method. We thank you for the opportunity to clarify the technical novelty of our work.
> We argue that the technical innovation of our contribution lies primarily in the following three aspects:
>
> **(1) Technical novelty**: Our proposed “paths-of-bifurcations” approximation is a novel approach, which raised several technical challenges:
> - Formulate the cost function on variables identified as needed and sufficient for an approximate variational inference to be able to recover ground truth parameters without accounting for the full tree structure;
> - Evaluation of quality of fits on real noisy binary tree-structured data observed in 3-D medical images;
> - Incorporate spatial constraints in the inference of HMM parameters;
> - Control optimization complexity with a limited number of model parameters;
>
> **(2) Increasing accessibility of complex methods**: We provide our code-implementation compatible with state-of-the-art sophisticated open-source packages, thus facilitating broader adoption of generative modeling for binary tree architectures.
>
> **(3) Robust validation on realistic synthetic data**: We include code to generate realistic airway trees based on previously-validated procedural models for readers to reproduce our results and further develop their own synthetic benchmarks (eg. to extend to botanics applications).
>
> **[R1-3] The method could be more clearly described in the main paper, and variables/distributions could be more clearly defined.**
>
> - The forms of HMM emission distributions, transition matrices, and start probabilities from Equations 1,2 are now detailed in Table 3 (Sup. B). We have also rephrased Section 2.1.1 (Generation of synthetic trees with hidden Markov modeling) for clarity.
>
> **[R1-4] "Fast and scalable inference of a generative process governing such tree-structures is key for classifying trees into categories." Are generative process modeling the prevalent approach to classification, or are there other (discriminative) approaches?**
>
> - For 3-D binary trees, discriminative approaches are prevalent. We have now emphasized this point in Section 1 (Introduction).
> - Our approach is the first to fit a generative model to a large, real dataset of 3-D human airway trees. We cite three other tree-HMM inference methods. However, none of these provide fast scalable open-source inference pipelines nor designed to be applicable beyond their specific use-case domain.
>
>
> [1] Mohammadi F, Visagan S, Gross SM, Karginov L, Lagarde JC, Heiser LM, Meyer AS. A lineage tree-based hidden Markov model quantifies cellular heterogeneity and plasticity. Communications Biology. 2022;5(1):1258.

---

> > ### Author Response · Authors · 2023-11-22
> > **Response to review of Submission6037 by Reviewer T8XB [Part 2/3]**
> >
> > **[R1-5] "... and split trees into 'independent' paths of bifurcations ..." why is independent in quotations?**
> > - In our "paths-of-bifurcations" approximation, we split each single binary tree into multiple paths. The term “independent” refers to the fact that while our approximation treats each path as independent, some paths are not truly independent as they come from the same tree.
> > - We validate this assumption via our synthetic benchmark experiments, in particular, in the low probability rules (LPR) experiments LPR-A, LPR-B (Section 3.3), where we test whether we can recover rare state rules in deep branch generations despite using overlapping paths that artificially over-sample non-rare early branching patterns.
> >
> >
> > **[R1-6] Would it be possible in the introduction to set up a more clear problem formulation so that is evident which types of data problem the paper addresses? For example, it is not clear whether the trees that are modelled have a graph-based or geometric representation, such as 3d coordinat-based or anatomical mesh representation.**
> >
> > - Our inference is designed for graph-based tree representations. We have added this distinction to Section 1 (Introduction).
> > - For our use-case on human airway trees, we included in the paper the steps required to go from geometric representations (segmentation masks) to graph representations (nodes and edges) in Sup. E.2.
> >
> >
> > **If I understand correctly, emission probabilities for branching angles and roll are Gaussian…**
> >
> > **[R1-7] Is anything done to ensure that the first angle "is the smallest angle by definition"**
> > - We confirm that we enforce that the first angle "is the smallest angle by definition"  to ensure that we do not infer redundant branching rules (e.g., where we would infer two distinct branching rules, while only different in the order of alpha1 and alpha2) when we observe an asymmetric bifurcation in our data (almost always).
> > - We have clarified this point in Section 2.1.1 (Generation of synthetic trees with hidden Markov modeling)
> >
> >
> > **[R1-8] …and that angles are within a suitable range?**
> > - During optimization angles are coded in radians ($-\pi$ to $\pi$), with $\alpha_1 < 0, \alpha_2 > 0$. But, we display angle values (in Tables 1-2, and Figures 2,3,6) and MAE (mean absolute error, Section 3) in degrees as this is more interpretable.
> > - We did not constrain the range of values on any angle variables.
> > - We have clarified these points in Section 2.3.2 (Training setup).
> >
> > **[R1-9] Would it be possible to provide some more details (possibly in the supplement) regarding the definitions of each term in the joint distribution (Eq. 1), such as a list with clear definitions of all variables and distributions.**
> >
> > - The forms of HMM emission distributions, transition matrices, and start probabilities from Equations 1,2 have been added in Table 3 (Sup. B).
> >
> > **[R1-10] Which variational distributions ('guide' in Pyro terminology) are used, and how important/sensitive is the choice?**
> > - We used the AutoDelta guide in Pyro and have added this information in Section 2.3.2 (Training setup).
> > - The complete model and inference codes are available from our GitHub link.
> >
> > **[R1-11] Have any other methods been used to model e.g. the human airways data? How does the proposed method perform in comparison?**
> >
> > - To our knowledge, we are the first group to propose a generative human airway tree model with extensive validation at individual level.
> > -  We mention previous procedural modeling methods for human airways in Section 3.5 (Human airway tree dataset from the ATM cohort).
> > - However, the ambition of all previous studies focused on documenting general population patterns/clusters while our goal is to extract subject-specific patterns. For example,  the previous well-cited method in (Wysoczanski  et al) [2]  compares human airway trees based on distances in BHV tree space with several pitfalls (tree trimming, complex computation, number of features increases exponentially with tree depth).
> >
> >
> >
> > [2] Wysoczanski A, Angelini ED, Smith BM, Hoffman EA, Hiura GT, Sun Y, Barr RG, Laine AF. Unsupervised clustering of airway tree structures on high-resolution CT: the MESA lung study. In Proceedings of International Symposium on Biomedical Imaging, 2021 (pp. 1568-1572).

---

> > > ### Author Response · Authors · 2023-11-22
> > > **Response to review of Submission6037 by Reviewer T8XB [Part 3/3]**
> > >
> > > **[R1-12] What would be the primary application of the proposed method?**
> > > - We are developing this method in the context of unsupervised discovery of new sub-phenotypes of human airway trees in the general population and in specific diseases (e.g., Chronic obstructive pulmonary disease, COPD). This is for us our primary application.
> > > - Sub-phenotyping within large cohorts enables to increase the power of downstream analysis on sub-populations of patients to: (1) formulate hypothesis on disease mechanisms, (2) detect biomarkers in specific sub-populations associated with individual susceptibility to disease, (3) create high-fidelity synthetic digital twins (here synthetic airway tree structures) for multiple applications such as particle deposition studies (deposition of smoke particles / environmental pollutants at bifurcation structures in the lung), association with genomic markers or inclusion in clinical trials.
> > >
> > > **[R1-13] Minor issues: There are a few typos etc. Examples from abstract and introduction: "...prevalent multiple across fields..."? Closing parenthesis doubled"))" in a few citations. "... and infer states..." should be 'and infers states' "... human airway trees segmented on CT images." Unclear**
> > > - We appreciate your attention to details when reviewing our paper and we have carefully proof-read the paper for typos.
> > >
> > > Once again, we sincerely thank the reviewer for the detailed, constructive comments and suggestions to improve our paper.
> > >
> > > Kind Regards,

---

> ### Comment · Reviewer_T8XB · 2023-11-23
> **Thank you**
>
> I appreciate your detailed responses. However, since my main concerns remain I will uphold my original score.

---

> ### Author Response · Authors · 2023-11-28
> **Request for feedback on our response**
>
> Dear Reviewer,
>
> We greatly appreciate your suggestions which have improved our work. As the author interaction is coming to an end (our extended deadline is 1st December), we are waiting for your post-rebuttal response. If our response answers your concerns, please consider raising your scores. If not, please let us know of your further questions such that we can answer them in the remaining time.
>
> Kind Regards,

---

> > ### Author Response · Authors · 2023-11-28
> > **Request to read ongoing discussion**
> >
> > Dear Reviewer,
> >
> > We are in discussions with Reviewer xwzK. We would greatly appreciate if you could take the time to read this discussion before you submit your final scores on our paper.
> >
> > Once again, many thanks for your suggestions and constructive feedback on our paper.
> >
> > Kind Regards,

---

### Official Review · Reviewer_xwzK · 2023-11-10

**Soundness:** 3 good
**Presentation:** 1 poor
**Contribution:** 1 poor
**Rating:** 5
**Confidence:** 5

**Summary:**

My sincere apologies, I posted the review of a different paper.

This paper is about HMMs on trees, and not about mutual information estimation. The authors use a variational technique to relax the problem, and introduce a parametric for states/transitions, making use of Categorical/Dirichlet and Gaussian/Inverse Gamma conjugacy.}

Updates: increased soundness score. Decreased presentation and contribution. The more I reread it the more I understand it to be simple if elegant modelling, which may not be appreciated by this community since, as a whole, ICLR is not a human biology conference.

**Strengths:**

-The authors address a interesting problem of expanding/branching HMMs.
-They use a standard solution to this problem.

**Weaknesses:**

* While I agree in spirit with the mathematical technique, I think section 2.3 needs work in clarifying notation (perhaps a block diagram?), and has possible errors due to its compression into a single page of 2 equations.
* It is not clear that air pathways actually have these complex state transitions? Could they instead be modeled using the (constant) branching process? Why are there hidden states, instead of constant branching factors or branching factors that are a function of depth? Basically, do the physical branching angles, rate of branching, etc., actually change?
* CT has imaging resolution on the order of mm (1mm isotropic is a very high quality human CT, multi-mm thick slice or gapped slices are standard clinical CT resolution). Do we expect the actual alveolar trees to be visible? If not, would these resolutions lead to non-tree like observations? (i.e., multi-compartment mergers)

**Questions:**

1) I'm fairly certain the ELBO should be a lower bound. Why can we have the log inside of the integral/summand in equation (1) without an inequality? Usually the latent variables are introduced at this stage, then either the spectral gap is derived, or Jensen's inequality used.
2) One motivating factor for this work is rule heterogeneity. However, is this actually possible using these priors? I think this is what your experiments are trying to show in 3.2; I encourage the authors to work on this. Branching processes with differing rules based on interesting conditions can be very interesting in applied fields, but Figures 4 and 5 are incredibly small and unreadable, as are table 1 and 2, but even more importantly, the authors should use the synthetic data to prove that if such a rule shift (so, a regime change between lung tissue structures) existed, it could be learned by this HMM, and wouldn't have been learned by the fixed branching process.

Additional notes: If the authors are set on using lung tissue, I feel a medical applied conference (MICCAI, MIDL, etc.) may be better suited for this type of contribution. While the HMM is interesting, its need within the application domain will not be appreciated by this community as much as a more medically inclined community.

---

> ### Author Response · Authors · 2023-11-22
> **Initial response to review of Submission6037 by Reviewer xwzK**
>
> We thank the reviewer for their detailed comments and suggestions.
>
> We are working on addressing your comments and questions with an extension granted till December 1st due to us only receiving the review yesterday.

---

> ### Author Response · Authors · 2023-11-27
> **Response to review of Submission6037 (Extended rebuttal deadline 1st Dec.) by Reviewer xwzK [Part 1/2]**
>
> We thank the reviewer for their detailed comments and suggestions.
>
> Thank you for recognizing our tractable solution to the interesting problem of branching HMM structures.
>
> We have prepared a detailed response below to your constructive comments and questions with our extended rebuttal deadline of 1st December.
>
> **[R3-1]  While I agree in spirit with the mathematical technique, I think section 2.3 needs work in clarifying notation (perhaps a block diagram?), and has possible errors due to its compression into a single page of 2 equations**
>
> - Many thanks for your attention to detail, we have corrected a typo in Equation 1.
> - The forms of HMM emission distributions, transition matrices, and start probabilities from Equations 1,2 have been added in Table 3 (Sup. B), along with line numbers corresponding to Algorithm 1 (Generative process) to increase clarity.
>
> **[R3-2] It is not clear that air pathways actually have these complex state transitions? Could they instead be modeled using the (constant) branching process? Why are there hidden states, instead of constant branching factors or branching factors that are a function of depth? Basically, do the physical branching angles, rate of branching, etc., actually change?**
>
> - Metzger et al., 2008 [1] hypothesizes and provides evidence to support lineage-based, genetically coded branching subroutines within the bronchial tree in animal models rather than a constant branching process. This formulation aligns with hidden Markov states rather than depth-based branching factors.
> - Sauret et al., 2002 [2] present the complex patterns of branching angles measured in two human bronchial trees. They report no significant difference in the plane rotation angle (= $\phi$, in our paper) with increasing branching generation. They highlight the wide distribution of all branching angles within branching generation=9 as an example. All the above contra-indicates a depth-based rule system for branching angles.
> - Figure 15 in Sup. E of our paper displays airway tree binary masks from the ATM dataset (used in Section 3.5). One can see the complex variation and patterning between subjects.
>
> **[R3-3] CT has imaging resolution on the order of mm (1mm isotropic is a very high quality human CT, multi-mm thick slice or gapped slices are standard clinical CT resolution). Do we expect the actual alveolar trees to be visible? If not, would these resolutions lead to non-tree like observations? (i.e., multi-compartment mergers)**
> - No, alveolar trees are not visible. We are working with clean airway binary segmentations with no alveolar trees,  as depicted in Figures 1c and 15.
>
> **[R3-4] I'm fairly certain the ELBO should be a lower bound. Why can we have the log inside of the integral/summand in equation (1) without an inequality? Usually the latent variables are introduced at this stage, then either the spectral gap is derived, or Jensen's inequality used.**
> - We agree with the properties of the ELBO that you described. However, Equation 1 is the log-joint and not the ELBO. In our work, the ELBO is derived automatically and numerically by Pyro.
> - We have chosen to detail the log joint (rather than the ELBO) in Equation 1 because from there, any optimisation method (e.g., expectation-maximization, variational inference) can be used to infer latent variables. In our GitHub repository, we demonstrate inference in Pyro and package ‘hmmlearn’, a Scikit-Learn extension package for Hidden Markov Models.
>
> **[R3-5] One motivating factor for this work is rule heterogeneity. However, is this actually possible using these priors? I think this is what your experiments are trying to show in 3.2; I encourage the authors to work on this.**
> - Yes, rule heterogeneity is what we are testing in Section 3.2. We have clarified the text and figures to demonstrate these results more clearly.
> - Please refer to Table 2 for the types of synthetic trees we validated our inference method on including  trees with rule heterogeneity, low probability rules and noisy data quality.
> We present the capabilities of our approximate inference method to recover HMM rules under these conditions in Section 3.
>
>
>
> **[R3-6] Branching processes with differing rules based on interesting conditions can be very interesting in applied fields, but Figures 4 and 5 are incredibly small and unreadable, as are table 1 and 2, but even more importantly,**
>
> - We have taken extensive measures to improve data presentation and summarization throughout the paper and ensure that all tables and figures are now readable at print size, especially in Section 3 (Results and Discussion).
>
>
>
> **References:**
>
> [1] Metzger RJ, Klein OD, Martin GR, Krasnow MA. The branching programme of mouse lung development. Nature. 2008;453(7196):745-50.
>
> [2] Sauret V, Halson PM, Brown IW, Fleming JS, Bailey AG. Study of the three-dimensional geometry of the central conducting airways in man using computed tomographic (CT) images. J Anat. 2002 ;200(Pt 2):123-34.

---

> > ### Comment · Reviewer_xwzK · 2023-11-28
> > **Response**
> >
> > Reading Metzger et al. 2008, the authors found that developmentally in mice there were three (or more) "subroutines": domain branching, in which branch buds emerge in alternating Lateral/Dorsal/Medial/Ventral directions (approximately 90 deg) from a single mother stem (the domain), and then planar and orthogonal branching, where bifurcation occurs either co-planar to the previous bifurcation or approximately 90 deg rotated.
> >
> > I am convinced that these events occur, and the relevance of the complex rule system, for which there are sets of latent states with specific patterning in their state-to-state transitions. I disagree with the authors' assertion that this is contra-indicative of a depth-based rule being sufficiently expressive: Metzger et al. 2008 Figure 3 sub-figure a clearly shows radial spatial patterning w.r.t. prox-distall axes. Metzger et al 2008 suggest that there is no fixed global order, which I think can be agreed viewing their results, but the degree to which these rules diverge from the results seems important as some quantification of the weight of evidence.
> >
> > I am also convinced however that this modelling, while clever, is better appreciated by the application domain community, or if the authors insist on generality, a modelling journal. The modelling is indeed clever for the application. It also offers little to the ICLR community. There is not an advancement in learning methods, there is simply pure application of HMM to this branching process. I admit, I was fooled by the enormity of the work, but, in my opinion, to the ICLR community, this is exactly an HMM on a tree-topology, and then an off-the-shelf solution. Such a statement is reductive, since it likely will show spatial and inter-subject heterogeneity and almost certainly not  uniformity w.r.t. the ruleset distributions, but that result is important for Lung and Lung Development sciences, not representation learning.
> >
> > This is not to say that this is a bad paper, but that it is purely modelling, and not research on the learning process.
> >
> > Specific notes:
> >
> > From the response:
> > > Equation 1 is the log-joint and not the ELBO.
> >
> > From the paper:
> > > For approximate inference of the HMM model detailed in Section 2.1.1, we optimize the objective
> > function in Equation 1, which involves the log joint distribution of the observed branching angles
> > and latent variables in the HMM (see Table 3, Sup. B for definitions of terms): [followed by the equation in question]
> >
> > It should be made clear that you are not optimizing Eq. 1, but a [mean field?] bound of it. Or Expectation Propagation or a higher order method.
> >
> > Looking at sections 2 and 3, I think the experiment outlined in Table 1 and Figure 2 sub-figure b (which is actually a table) appears to be presenting what I asked for, but they are visually disjoint, and could be presented as plots (similar to Figure 2 sub-figure a, which has three sub-sub-figures.). Present these as 3 large, legible figures, with two colors of Gaussian density (recalling the $\alpha_0, \alpha_1, \varphi$ distributions are fully disentangled as the joint pdf is product of the marginals) for each of the rule sets, instead of plotting both ruleset on the same set of axes. While Figure 8 and Figure 9 are interesting when viewed as a whole, they plot these values (estimated and ground-truth) on separate axes, but retain the rulesets on the same axes.
> >
> > I appreciate that the authors have attempted to edit their tables and figures, but I am only able to read Figures 2, 3, and 4 under 200% magnification, and even then I need a second window open at 100% magnification to read the context of the figures in the text. Perhaps I am old, but this seems to be the source of the reviewers' qualms about readability.
> >
> > Figure 6 to me seems like at once the most interesting, and the least meaningful, which is why I suggest that this is misplaced in an ML conference. It feels like subfigure a) should be much larger, and that the substates from models using 7-12 hidden states should be assessed using Saherian & Yoon 2011's framework (using a symmetric divergence and not KL). Do you actually find additional states as you increase past 6? Or are we overfitting by sub-dividing modes into very similar sub-modes, and only receiving a marginal log(n+k) penalty from BIC for the extra parameters? Are there minority modes that are hiding in k=7...? This seems scientifically relevant, but somewhat irrelevant to the ML community.
> >
> > From the response:
> > > We are working with clean airway binary segmentations with no alveolar trees, as depicted in Figures 1c and 15.
> >
> > Great, are these always strictly trees or do partial voluming/inter-slice spacing effects cause merger of branches into non-tree structures?
> >
> > Overall, I'm convinced that it works, and that the rule heterogeneity is being modelled at the level of the latent states themselves. But now, after reading it more thoroughly, I'm left wondering why this should be published in a ML venue.

---

> > > ### Author Response · Authors · 2023-11-28
> > > **Response to follow up comments on Submission6037 by Reviewer xwzK**
> > >
> > > Dear Reviewer,
> > >
> > > Thank you for sharing your thoughts. We think that almost your entire response is based on your opinion as to “Why ICLR'' rather than on the high quality our work and its wide-ranging applicability in different fields of science. We would like to point out that the suitability of our research was not questioned by the other two reviewers.
> > >
> > > Our paper is on efficient **L**earning of **R**epresentations (...the LR in ICLR) for 3-D binary tree structures with a novel approximation (“paths-of-bifurcations”) during inference and control of parameters. We considerably simplify the upfront parameterisation of such tree-structures to enable reliable inference, relying on specialized open-source libraries, accessible to a large community of researchers in need of such models. Given our focus on Learning Representations, we sincerely believe that ICLR is the exact place to present our research, and not a medical conference.
> > >
> > > **Further, we already detailed our arguments on the scope of our paper in our previous response to your comment in [R3-7]** (https://openreview.net/forum?id=YTKShuSOhI&noteId=sy3ctBnaGJ).
> > >
> > > *Addressing some more specific points below:*
> > >
> > > **[R3v2-1] It should be made clear that you are not optimizing Eq. 1, but a [mean field?] bound of it**
> > >
> > > You are correct, we were not specific enough, we fix this in the latest version of our paper. Thank you for the suggestion.
> > >
> > > **[R3v2 -2] Present these as 3 large, legible figures, with two colors of Gaussian density ...**
> > >
> > > This comment relates to Figure 2a-b.
> > >
> > > 1. For space constraints we present our inferred parameters in a table in Figure 2b rather than 3 figures.
> > > 2. The baseline inference quality is excellent, as can be seen from the table in Figure 2b. The inferred emission distributions agree with the ground truth to within a mean absolute error MAE < 0.2 degrees (maximum difference = 0.5 degrees). This difference between the inferred emission distributions and the ground truth (shown in Figure 2a) is too small to be distinguished via pictures of the distributions.
> > >
> > >
> > > **[R3v2-3] While Figure 8 and Figure 9 are interesting when viewed as a whole, they plot these values (estimated and ground-truth) on separate axes, but retain the rulesets on the same axes.**
> > >
> > > Incorrect. Figures 8 and 9 do not contain any estimated values. They are visual displays of the emission distributions we tested in RH-A, RH-B experiments, i.e it is a visualization of the ‘modified’ ground truth with parameters in Table 2.
> > >
> > > **[R3v2-4] I am only able to read Figures 2, 3, and 4 under 200% magnification, and even then I need a second window open at 100% magnification to read the context of the figures in the text...**
> > >
> > > We beg to differ. Please see the link below to a few example ICLR 2023 papers with similar Figure / Text size. We relied on previous ICLR papers such as these while preparing our revised figures.
> > >
> > > Example 1:  Figure 1 in Huang F, Lu K, Yuxi CA, Qin Z, Fang Y, Tian G, Li G. Encoding Recurrence into Transformers. (https://openreview.net/forum?id=7YfHla7IxBJ)
> > >
> > > Example 2: Figure 3 in Hron J, Krauth K, Jordan MI, Kilbertus N, Dean S. Modeling content creator incentives on algorithm-curated platforms. (https://openreview.net/forum?id=l6CpxixmUg)
> > >
> > >
> > > **[R3v2-5] Figure 6 to me seems like at once the most interesting, and the least meaningful ,which is why I suggest that this is misplaced in an ML conference. It feels like subfigure a) should be much larger,.. Are there minority modes that are hiding in k=7...?.**
> > >
> > > We strongly disagree on the importance of an example use-case in our paper which is why Figure 6 is smaller. Figure 6 is not driving where we want to publish.
> > >
> > > We do not include details on the sub-modes etc., because, as you mention, these results would be of interest in a medical conference paper not a methodology-focused paper.
> > >
> > > Our paper focuses on validating the ‘paths-of-bifurcations’ approximate inference method on synthetic 3-D trees with space constraints, rule heterogeneity, low probability rules and noisy data quality (Figures 2-5). **[cf. our responses to ‘Why ICLR’]**.
> > > This is definitely not a paper suitable for the MICCAI / MIDL kind of conferences you suggest. We would like to emphasize once again that the medical application is NOT the focus of our paper, it is simply a use-case to add a dimension to the complexity of the binary trees we need to handle.
> > >
> > > Our synthetic data is based on a real-case scenario with complexity of binary trees that a community of researchers in wide ranging fields would be interested in handling, such as airways, vessels, cell lineages, botanical trees to name a few. We believe that the simplicity and versatility of our “paths-of-bifurcations” inference approach will find its applicability in many fields other than medical.
> > >
> > > We thank the reviewer for their feedback, and respectfully present our responses regarding your opinion on the scope of ICLR both in this response and in our previous.
> > >
> > > Kind Regards,

---

> ### Author Response · Authors · 2023-11-27
> **Response to review of Submission6037 (Extended rebuttal deadline 1st Dec.) by Reviewer xwzK [Part 2/2]**
>
> **[R3-7] the authors should use the synthetic data to prove that if such a rule shift (so, a regime change between lung tissue structures) existed, it could be learned by this HMM, and wouldn't have been learned by the fixed branching process.**
>
> - We do use extensive synthetic data in Section 3.2, Sup. D.2
> - We demonstrate that an oracle classifier (which knows about non-constant branching rules) underperforms on such structures without lineage information. The oracle we show is already a more advanced classifier than a fixed rule model.
>
> **[R3-7] Additional notes: If the authors are set on using lung tissue, I feel a medical applied conference (MICCAI, MIDL, etc.) may be better suited for this type of contribution. While the HMM is interesting, its need within the application domain will not be appreciated by this community as much as a more medically inclined community.**
>
> - We want to publish this work in ICLR as it has potentials for several applications (e.g., botanics, genomics, wavelet methods).
>
> - In addition, the content goes beyond MICCAI in terms of methodological formulation and validation on synthetic airway tree data, while it does not yet address the clinical value of our modeling for downstream tasks (e.g., diagnosis, clinical associations of symptoms within clusters) which is typically expected in a MICCAI paper. Additionally, we are only marginally related to medical image computing as the CT scans are already segmented. We are only studying the shape of binary segmentation masks.
>
> - MIDL is not appropriate as this is not a Deep Learning based framework.
> - This is a methodology focussed paper rather than a clinical paper, with an example application in healthcare, which is within the scope of ICLR.
>
>
> Once again, we sincerely thank the reviewer for the detailed, constructive comments and suggestions to improve our paper.
>
> Kind Regards,

---

### Author Response · Authors · 2023-11-22
**Response to all reviewers**

Dear Reviewers,

Many thanks for your insightful comments, constructive suggestions and support of our paper.

We have responded in detail to each comment made and indicated sections of the revised document where changes can be found.

We have also, as a supplementary material pdf, uploaded a version of the main document, coloured for changes for ease of review.

The version in the main pdf, and in the supplement, are identical save for colouring of revised text / figures in blue in the supplemental pdf.

Please let us know if there are any issues, and thank you all once more for the detailed reviews.

Kind Regards,

---

### Meta-Review · Area_Chair_WMDY · 2023-12-06

**Metareview:**

This paper explores a paths-of-bifurcations inference approach to modeling tree structures with HMMs. The method seems to be sound and interesting contributions. However, most of reviewers feel that the paper is not easy to read. In addition, they also feel that this paper is perhaps better appreciated in other venues (not ICLR). Even after the author response and during the AC-reviewer discussion period, they kept their initial decisions. I understand that the authors made efforts in responding to many concerns raised by reviewers. However, we feel that the paper is not ready for being published on ICLR in its current version. Therefore, the paper is not recommended for acceptance in its current form. I hope authors found the review comments informative and can improve their paper by addressing these carefully in future submissions.

**Justification For Why Not Higher Score:**

This paper is perhaps better appreciated in other venues (not ICLR).

**Justification For Why Not Lower Score:**

N/A

---

### Decision · Program_Chairs · 2024-01-16

Reject